# TOWARDS REALISTIC UNSUPERVISED FINE-TUNING WITH VISION-LANGUAGE MODELS

## ABSTRACT

The emergence of vision-language models (VLMs), such as CLIP, has spurred a significant research effort towards their application for downstream supervised learning tasks. Although some previous studies have explored the unsupervised fine-tuning of CLIP, they often rely on prior knowledge in the form of class names associated with ground truth labels. In this paper, we delve into a realistic unsupervised fine-tuning scenario by assuming that the unlabeled data might contain out-of-distribution samples from unknown classes. Furthermore, we emphasize the importance of simultaneously enhancing out-of-distribution detection capabilities alongside the recognition of instances associated with predefined class labels.

To tackle this problem, we present a simple, efficient, and effective fine-tuning approach called Universal Entropy Optimization (UEO). UEO leverages sample-level confidence to approximately minimize the conditional entropy of confident instances and maximize the marginal entropy of less confident instances. Apart from optimizing the textual prompts, UEO also incorporates optimization of channel-wise affine transformations within the visual branch of CLIP. Through extensive experiments conducted across 15 domains and 4 different types of prior knowledge, we demonstrate that UEO surpasses baseline methods in terms of both generalization and out-of-distribution detection.

## 1 INTRODUCTION

Vision-language models (VLMs) (Radford et al., 2021; Li et al., 2022a; Jia et al., 2021; Li et al., 2022c) pre-trained on web-scale image-text pairs have exhibited robust zero-shot prediction capabilities, which have recently attracted increasing attention from the research community. As an example, Contrastive Language-Image Pretraining (CLIP) (Radford et al., 2021) leverages a contrastive objective to obtain a modality-agnostic embedding space in which the paired images and texts are pulled closer and unpaired images and texts are pushed apart. Subsequently, CLIP can perform zero-shot visual prediction by matching the embeddings of test images and prompt-based textual descriptions (*e.g.*, "a photo of a [CLASS]" and "this is a picture of a [CLASS]"), merely requiring the names of all the semantic classes in downstream tasks.

Apart from the extensive research dedicated to the pre-training stage, numerous studies (Zhou et al., 2022b; Zhang et al., 2022b; Bahng et al., 2022) have concentrated on adapting VLMs to specific downstream tasks by using task-specific labeled data. This fine-tuning paradigm empowers VLMs to bridge both data and task gaps, leading to improved performance in recognition tasks. In addition to multi-class classification, these pioneering strategies have also been harnessed in a spectrum of computer vision tasks, including ordinal regression (Li et al., 2022b), point cloud understanding (Zhang et al., 2022a), and dense prediction (Rao et al., 2022). When considering fine-tuning setups, most efforts have primarily revolved around fully supervised and few-shot supervised learning scenarios. To pursue annotation efficiency and scalability, several recent studies (Huang et al., 2022; Shu et al., 2022; Tanwisuth et al., 2023) have delved into the realm of unsupervised fine-tuning for VLMs, remarkably achieving performance on par with few-shot supervised approaches. However, they still demand a priori knowledge of class names associated with ground truth labels, limiting their applicability in diverse real-world scenarios.

To circumvent the limitation, this paper explores a novel fine-tuning setup, termed Unsupervised Universal Fine-Tuning (U$^2$-FT) with VLMs, wherein the predefined list of class names may partially

overlap with the ground truth label space of unlabeled training data. To illustrate, as depicted in Fig. 1, consider an unlabeled data set **[C2]** comprising samples from three classes (*i.e.*, 'dog', 'cat', and 'panda'). However, the provided predefined list of class names might be imprecise, containing four classes (*i.e.*, 'fox', 'dog', 'cat', 'pig'). Generally, $U^2$-FT requires the fine-tuned model to demonstrate superior performance to the original VLMs in two aspects, namely, recognizing samples from classes within the predefined list ('dog' and 'cat'), as well as identifying samples from classes not present in that list ('panda'), commonly referred to as out-of-distribution (OOD) samples. Given the potential scarcity of OOD samples **[C1]**, $U^2$-FT evaluates both generalization and OOD detection through a new test data set encompassing both OOD samples and samples from classes within the predefined list. Typically, $U^2$-FT poses two primary technical challenges for designing fine-tuning strategies: (1) fitting the entire data with VLMs will deteriorate the ability to detect OOD samples due to the potential presence of OOD samples, and (2) matching the label distribution of unlabeled data with the pre-defined one can be risky due to the potential absence of certain classes.

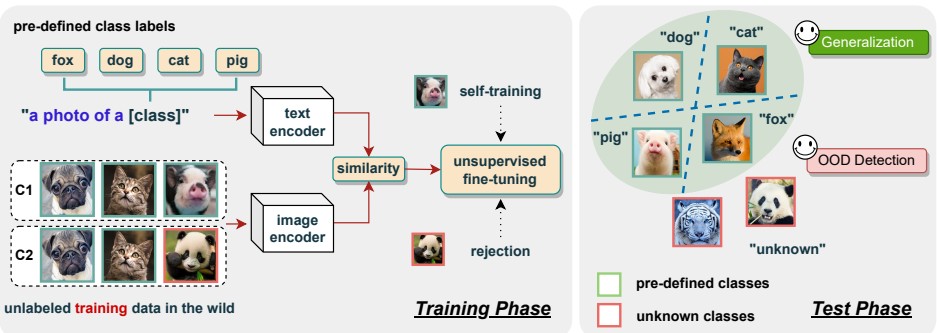

Figure 1: The basic setup of Unsupervised Universal Fine-Tuning ($U^2$-FT). During the training phase, $U^2$-FT fine-tunes the pre-trained VLMs with unlabeled in-the-wild training data according to an imprecise predefined list of class names (where 'fox' may be absent and 'panda' may be included). A new data set in the test phase is employed to evaluate performance across both generalization and OOD detection aspects.

We propose to address the challenges by presenting a parameter-efficient approach termed Universal Entropy Optimization (UEO). UEO aims to minimize the information entropy of non-OOD samples while simultaneously maximizing the entropy of OOD samples. Since we do not know which samples are OOD, UEO readily utilizes the confidence of unlabeled data in VLMs as sample-level weight. To avoid the potential risks associated with OOD sample exposure through entropy maximization, UEO employs a reverse weighting strategy to aggregate the predictions first, before subsequently maximizing the marginal entropy. Besides, UEO takes into account the optimization of channel-wise affine transformations in the image encoder of CLIP, in addition to the textual prompts, to ensure parameter efficiency. Overall, UEO is remarkably simple, requiring alterations to only a few lines of code. Our contributions are summarized as follows: (1). We introduce a new unsupervised fine-tuning setup with VLMs that requires minimal prior knowledge of the label space for unlabeled data. (2). Alongside achieving enhanced generalization performance, we simultaneously investigate the efficacy of fine-tuning VLMs for OOD detection. (3). We propose a new parameter-efficient approach, UEO, which elegantly incorporates the sample-level confidence during entropy optimization for unlabeled data in the wild. (4). Through extensive experiments, we demonstrate that UEO consistently outperforms existing methods across 15 diverse downstream domains.

## 2 RELATED WORK

### 2.1 PROMPT TUNING OF VLMs

Beyond a variety of investigations on pre-training VLMs (Radford et al., 2021; Jia et al., 2021; Yao et al., 2022), one line of research explores VLMs with transfer learning to improve their ability to generalize to specific downstream tasks (Zhang et al., 2023). Prompt tuning (Lester et al., 2021), initially devised for adapting language models, optimizes the text embedding space while leaving

the foundational model parameters unchanged. This technique has gained significant interest in fine-tuning VLMs for vision tasks (Zhou et al., 2022a;b; Zhu et al., 2023) owing to its parameter-efficient nature. For example, CoOp (Zhou et al., 2022b) employs prompt tuning within the language branch of CLIP by utilizing a limited amount of labeled data, while VPT (Bahng et al., 2022) introduces visual prompts by modifying the pixels of images. To offer greater versatility, both UPT (Zang et al., 2022) and MaPLE (Khattak et al., 2023) propose to optimize prompts within both the vision and language branches.

Depending on the availability of annotations during the learning process, current fine-tuning setups of VLMs fall within three main categories: fully supervised transfer (Rao et al., 2022; Bahng et al., 2022; Wortsman et al., 2022), few-shot supervised transfer (Zhou et al., 2022b;a; Zhang et al., 2022b), and unsupervised transfer (Huang et al., 2022; Shu et al., 2022; Li et al., 2023; Tanwisuth et al., 2023). Unlike the other two transfer setups that depend on labeled downstream data, unsupervised transfer only harnesses unlabelled downstream data for fine-tuning VLMs. This paradigm sounds more challenging but also more promising and efficient for adapting VLMs, as it does not require costly annotation efforts and can better capture the underlying data distribution. This paper extends the scope of existing unsupervised transfer methods by assuming that not all unlabeled data belongs to classes within a predefined list.

## 2.2 OOD Detection of VLMs

Out-of-distribution (OOD) detection (Yang et al., 2021) focuses on identifying instances or examples unrelated to the in-distribution task, which is critical for the real-world deployment of machine learning models. However, exploring VLMs for OOD detection remains an interesting but relatively recent research topic, with only a handful of previous efforts within this field (Fort et al., 2021; Esmaeilpour et al., 2022; Ming et al., 2022; Wang et al., 2023). Unlike (Fort et al., 2021; Esmaeilpour et al., 2022), which rely on prior information on OOD samples, MCM (Ming et al., 2022) offers a training-free OOD detection approach that measures the similarities between visual features with textual concepts. Besides, a recent study (Liao et al., 2023) fine-tunes VLMs using labeled data to improve both generalization and outlier detection performance, even incorporating novel words from WordNet. Two concurrent studies (Ming & Li, 2023; Miyai et al., 2023) further investigate the performance of fine-tuned CLIP after few-shot in-distribution (ID) classification. In contrast, our approach enhances the OOD detection performance of VLMs using only unlabeled data, without any additional information apart from a predefined list of class names.

## 2.3 Unsupervised Model Adaptation

Unsupervised model adaptation (*a.k.a.*, source-free domain adaptation) (Liang et al., 2020; Li et al., 2020; Huang et al., 2021; Kundu et al., 2022) has emerged as a popular research topic in transfer learning, with the goal of transferring a well-trained model from a labeled source domain to an unlabeled but related target domain. As classified in a recent survey paper (Liang et al., 2023), unsupervised model adaptation methods can be broadly categorized into four popular schemes: pseudo-labeling, consistency training, clustering-based training, and source distribution estimation. However, these methods typically require a closely related source domain to train the source model, which may limit their practical applicability in real-world scenarios. Conversely, by employing VLMs such as CLIP, we can effortlessly acquire a high-quality source model with the help of class names from the target task.

While most research in unsupervised model adaptation has focused on closed-set scenarios, a few studies (Liang et al., 2021; Feng et al., 2021; Qu et al., 2023) have also investigated open-set model adaptation, which deals with target tasks that contain additional classes not present in the source task. In such cases, these novel classes are treated as a distinct 'unknown' category, and their accuracy is evaluated accordingly. Nonetheless, existing model adaptation techniques are always tailored to address a single category shift scenario, specifically, open-set transfer (Kundu et al., 2020; Feng et al., 2021; Liang et al., 2021; Qu et al., 2023) or closed-set transfer (Liang et al., 2020; Li et al., 2020; Huang et al., 2021; Liang et al., 2022). To the best of our knowledge, DANCE (Saito et al., 2020) is currently the only domain adaptation method that can handle different types of category shift scenarios (*i.e.*, closed-set, partial-set, open-set, and open-partial-set) without prior knowledge of the specific scenario. Building upon the universal domain adaptation concept of DANCE, this

paper presents the problem of unsupervised universal fine-tuning with VLMs, which adapts CLIP to unlabeled training data with the presence of an imprecise predefined list of class names. It is worth noting that we employ OOD detection to distinguish between known classes and OOD classes, thereby avoiding the need for sensitivity thresholds.

# 3 UNSUPERVISED UNIVERSAL FINE-TUNING WITH CLIP

## 3.1 PRELIMINARY

In this paper, we employ CLIP (Radford et al., 2021) as a representative VLM for unsupervised universal fine-tuning throughout this paper since it is a pioneering work that has led to significant advancements in various computer vision tasks. For the sake of simplicity, we focus on the image classification task. Typically, the CLIP model adopts a straightforward dual-stream architecture with an image encoder, denoted as $g_I(\cdot)$, and a text encoder, denoted as $g_T(\cdot)$. Each encoder processes input data from the corresponding modality. During its pre-training phase, CLIP leverages a self-supervised contrastive objective to learn image-text correspondences from noisy pairs of images and text sourced from the Internet. As a result, the features of paired images and texts are close to each other in the shared embedding space.

To facilitate zero-shot prediction in downstream tasks, CLIP generates a prompt (*e.g.*, "a photo of a [CLASS]") for each class by replacing the [CLASS] token with the name of the corresponding class. This technique aims to reduce the gap between the text distribution of the pre-training dataset and that of the target downstream task. Then, we get the text embedding of each class encoded by the text encoder $\{\mathbf{T}_c\}_{c=1}^C$, where $C$ denotes the number of classes in the target task. For making predictions, we compare the image embedding $\mathbf{I}_x = g_I(x)$ of an input image $x$ against a set of text embeddings $\{\mathbf{T}_c\}_{c=1}^C$ and obtain the probability that $x$ belongs to class $c$ using a softmax operation:

$$p_c(x) = p(y = c|x) = \frac{\exp(\mathbb{S}(\mathbf{I}_x, \mathbf{T}_c)/\tau)}{\sum_{i=1}^C \exp(\mathbb{S}(\mathbf{I}_x, \mathbf{T}_i)/\tau)}, \tag{1}$$

where $\mathbb{S}(\cdot, \cdot)$ denotes the cosine similarity metric between embeddings, and the temperature parameter $\tau$ is set to 0.01 by default. Note that the accuracy of zero-shot inference is highly dependent on the quality of the candidate class names $\{y_1, \ldots, y_C\}$ selected for the prediction task.

In addition to its impressive zero-shot classification capabilities, CLIP has also demonstrated remarkable performance in zero-shot OOD detection as reported in (Ming et al., 2022; Wang et al., 2023). For the sake of simplicity, we adopt the maximum softmax probability score as,

$$S(x) = \max_c p_c(x). \tag{2}$$

Due to the strong zero-shot classification ability, ID samples will be matched to one of the textual descriptions in the candidate list with a high score, and vice versa. Formally, a standard OOD detection function can be expressed as: $f_\lambda(x) = \begin{cases} \text{ID} & S(x) \geq \lambda \\ \text{OOD} & S(x) < \lambda \end{cases}$, where $\lambda$ is a chosen threshold so that a high fraction of ID data is above the threshold in real-world applications. For samples that are categorized as ID, we easily obtain the class prediction through $\hat{y} = \arg\max_c p_c(x)$.

## 3.2 PROBLEM SETTING

With a predetermined name list of interested classes, denoted as $\{y_1, \ldots, y_C\}$, the objective of Unsupervised Universal Fine-Tuning ($\text{U}^2$-FT) is to facilitate the adaptation of pre-trained VLMs to unlabeled data $\mathcal{X}_t$ in the wild. $\text{U}^2$-FT is primarily designed to enhance the performance of VLMs in two key aspects: (1) accurately classifying samples affiliated with the 'known' classes from the aforementioned list (ID generalization) and (2) effectively identifying samples beyond these designated classes (OOD detection). To better understand the wildness of unlabeled data, we denote the label space of the predefined list as $L_p$, and the label space of the unlabeled data as $L_u$. Prior unsupervised fine-tuning methods (Huang et al., 2022; Tanwisuth et al., 2023) only consider the closed-set category shift scenario (*i.e.*, $L_u = L_p$). However, three other prevalent category shift scenarios exist: partial-set ($L_u \subset L_p$), open-set ($L_p \subset L_u$), and open-partial ($L_u \cap L_p \neq \emptyset, L_u \not\subset L_p, L_p \not\subset L_u$).

Due to the unlabeled nature of downstream data, we may not know in advance which of these scenarios will occur. Hence, we adhere to the notion of 'universal' as introduced in the pioneering work of DANCE (Saito et al., 2020), delving into the universal adaptation of VLMs to wild unlabeled data.

**Evaluation.** Generally, we evaluate the recognition performance after unsupervised fine-tuning across four distinct category shift scenarios. Unlike DANCE (Saito et al., 2020), which assesses accuracies using training unlabeled data, we opt for an independent test set comprising both ID and OOD samples during evaluation. This choice is motivated by the fact that, in scenarios involving closed-set and partial-set category shifts, the absence of OOD samples in the test set makes it unfeasible to assess the performance of OOD detection. Furthermore, DANCE treats all OOD samples as an extra 'unknown' class and subsequently calculates the OS score (Panareda Busto & Gall, 2017; Saito et al., 2018), which represents the accuracy averaged over all classes. More recently, numerous open-set domain adaptation methods have embraced the HOS score (Bucci et al., 2020; Fu et al., 2020), replacing the simple average with the harmonic mean.

Both OS and HOS scores necessitate the accuracy of the 'unknown' class, which highly relies on the selected threshold $\lambda$. As an alternative approach, we align with the prevalent practice in the field of OOD detection (Yang et al., 2021; 2023) and incorporate another widely accepted metric: the area under the receiver operating characteristic curve (AUC). This metric is complemented by the assessment of per-class accuracy for ID samples within the test set. In Fig. 2(a-b), we first illustrate the sensitivity of OS and HOS scores of different methods as the threshold varies. Clearly, the highest OS score is consistently achieved when no samples are classified as OOD. The variation in HOS score differs across different methods and is notably affected by the threshold. In addition, as depicted in Fig. 2(c), the examination reveals a positive linear correlation between the AUC score and the maximum HOS score, indicating the superiority of the AUC score for open-set evaluation.

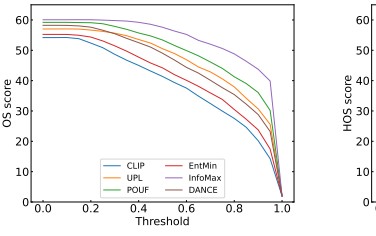 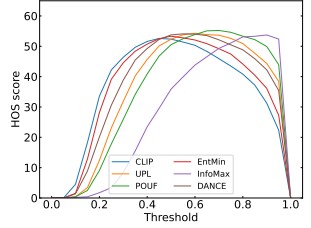 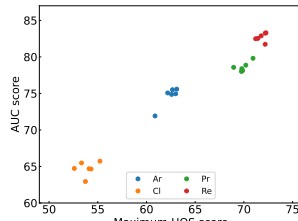

(a) OS under different thresholds    (b) HOS under different thresholds    (c) AUC v.s. the maximum HOS score

Figure 2: OS and HOS scores of different methods with the change of threshold under open-partial category shift on the Cl domain of OfficeHome (Venkateswara et al., 2017) are shown in (a-b). The relationship between AUC and the maximum HOS score is depicted in (c) for different domains.

### 3.3 UNIVERSAL ENTROPY OPTIMIZATION (UEO)

To adapt CLIP to the unlabeled data, we consider using Shannon entropy as the optimization objective function. The idea is to minimize the entropy of the model's prediction for each instance, making them closer to one of these prototypes in the feature space. However, when the training data contains OOD samples, entropy minimization may have the unintended effect of weakening the model's ability to reject them. Ideally, OOD samples should exhibit dissimilarity to any of the classes in the predefined list, thereby we can use entropy maximization instead to make the model produce approximately uniform predictions. Nevertheless, this approach becomes infeasible in unsupervised fine-tuning scenarios due to the lack of knowledge about which samples are OOD.

Previous studies (Ming et al., 2022; Wang et al., 2023) have shown that using the maximum softmax probability score in Eq. (2) for CLIP can achieve impressive performance in OOD detection. Inspired by this, we treat such scores as sample-level weights $w(x)$ to approximately achieve entropy minimization and maximization at the same time. Formally, the unified objective of entropy optimization is written as,

$$\mathcal{L} = \sum_{x \in \mathcal{B}_t} \widetilde{w}(x)\mathcal{H}(p(x)) - \sum_{x \in \mathcal{B}_t} \widetilde{\Phi}(w(x))\mathcal{H}(p(x)), \tag{3}$$

where $\mathcal{B}_t$ denotes a mini-batch of $\mathcal{X}_t$, $\mathcal{H}(p(x)) = -\sum_{c=1}^{C} p_c(x)\log p_c(x)$ denotes the Shannon entropy of $p(x)$, and $\Phi(\cdot)$ represents a monotonically decreasing function, such as $\Phi(w) = 1/w$. The

normalized weight within a mini-batch is defined as $\widetilde{w}(x) = \frac{w(x)}{\sum_x w(x)}$, which emphasizes confident samples during entropy minimization. In contrast, the normalized weight before entropy maximization is denoted as $\widetilde{\Phi}(w(x)) = \frac{\Phi(w(x))}{\sum_x \Phi(w(x))}$, which places emphasis on potential OOD samples.

The combination of two weighted entropy terms in Eq. (3) may be optimal when the weights within a mini-batch exhibit significant diversity. If no OOD samples are present in the mini-batch, the second entropy term would deteriorate the adaptation process by increasing the entropy of a difficult sample belonging to one of the targeted classes. To mitigate these potential risks, we apply entropy maximization over the average prediction of all the OOD samples and obtain the following objective,

$$\mathcal{L} = \sum_{x \in \mathcal{B}_t} \widetilde{w}(x)\mathcal{H}(p(x)) - \mathcal{H}(\bar{p}), \text{ where } \bar{p} = \sum_{x \in \mathcal{B}_t} \widetilde{\Phi}(w(x))p(x) \tag{4}$$

is the weighted average of predictions for each sample within the mini-batch $\mathcal{B}_t$. Intuitively, the objective in Eq. (3) resembles the mutual information maximization loss (Liang et al., 2020), which can also be decomposed into two entropy terms. When all the samples within a mini-batch share the same weight, namely, $\widetilde{w}(x) = \widetilde{\Phi}(w(x)) = \frac{1}{\|\mathcal{B}_t\|}$, where $\|\mathcal{B}_t\|$ denotes the batch size, the objective exactly degrades to the information maximization loss. This indicates that even if no OOD samples exist in the unlabeled data, optimizing the second term in Eq. (4) can still be beneficial.

**Parameter efficiency.** During the unsupervised adaptation process, we employ a parameter-efficient fine-tuning paradigm (Lialin et al., 2023) for foundation models, wherein only a small number of parameters instead of the entire model are modified during the fine-tuning process. In particular, we follow CoOp (Zhou et al., 2022b) by optimizing the text prompts, namely the learnable word vectors $\{[V_i]\}_{i=1}^m$ in the textual sentence "$[V_1], [V_2], \ldots, [V_m], [CLASS]$", where $m$ denotes the length of text prompts. Several prior studies (Bahng et al., 2022; Zang et al., 2022; Khattak et al., 2023) have demonstrated that the integration of visual prompts enhances the few-shot labeled adaptation of CLIP. However, it is worth noting that these methods are exclusively suitable for transformer-based visual branches. In this study, we draw inspiration from TENT (Wang et al., 2021) and introduce an approach to optimize the affine parameters within the normalization layers of the image branch, complementing the optimization of text prompts.

## 4 EXPERIMENTS

### 4.1 SETUP

**Datasets.** We validate $U^2$-FT methods on four widely used domain adaptation datasets, *i.e.*, **Office (OF)** (Saenko et al., 2010) that comprises 3 domains of 31 object categories, **OfficeHome (OH)** (Venkateswara et al., 2017) that encompasses 65 categories across 4 domains, **VISDA-C (VD)** (Peng et al., 2017) that includes 2 distant domains of 12 classes, and **DomainNet (DN)** (Peng et al., 2019) that contains 345 classes distributed in 6 styles.

**Protocols & Evaluations.** $U^2$-FT assesses both the ID generalization performance of interested classes from a predefined list and the OOD detection capability towards samples from OOD classes, as described in Section 3.2. Thus, we split the label space of the downstream task into two folds: interested classes and outlier classes. It is worth noting that the test set is kept unchanged across all four different category shift scenarios. To evaluate the performance of UEO under different category shifts, we determine the label space of the training unlabeled data based on the specific shift and select all the data from these classes. The detailed data splits for each dataset and specific shift can be found in Appendix A.2. Concerning the ID generalization performance, we use the per-class accuracy metric (ACC). On the other hand, to evaluate outlier detection ability, we use maximum softmax probability as the confidence for all the samples and measure the AUC score.

**Baseline methods.** We conduct a comparative analysis of UEO against several unsupervised fine-tuning methods, including UPL (Huang et al., 2022) and POUF (Tanwisuth et al., 2023), as well as the robust zero-shot inference baseline, CLIP (Radford et al., 2021). Besides, we present the results of one modified domain adaptation method, DANCE (Saenko et al., 2010), and two model adaptation methods (*i.e.*, Entropy Minimization (EntMin) (Wang et al., 2021) and Mutual Information Maximization (InfoMax) (Liang et al., 2020)). In DANCE, the source loss is omitted, and we set the trade-off before two target losses as 0.1. Furthermore, we introduce an oracle method, labeled

as UEO(O), wherein the weight $w(x)$ in Eq. (4) is equal to 1 if it belongs to one of the classes in the predefined list and 0 otherwise.

**Implementation details.** For all experiments, we utilize the pre-trained ResNet-50 and ViT-B/16 models provided by the official CLIP repository (Radford et al., 2021). The epoch number is set to 50 for small-size datasets (*i.e.*, Office and OfficeHome) and 5 for large-size datasets (*i.e.*, VISDA-C and DomainNet), and the learned model in the last epoch is chosen for a fair evaluation. During training, we use an SGD optimizer with an initial learning rate of 1e-4 for both encoders, except for EntMin, which uses a learning rate of 1e-5. We also employ a cosine scheduler to gradually decrease the learning rate. The parameters optimized in all methods include the prompt of the text encoder and affine parameters in the normalization layers (*i.e.*, BatchNorm in ResNet-50 and LayerNorm in ViT-B/16) of the visual encoder. The context length of the prompt is fixed at 4 and takes the default initialization "a photo of a". We reproduce all methods' loss functions using the hyperparameters provided in their respective papers with the experimental results averaged over two different seeds.

## 4.2 EXPERIMENTAL RESULTS

We evaluate the UEO method for various scenarios of U²-FT, and also reproduce the baseline methods for comparison. The results under four distinct category shift scenarios (*i.e.*, closed-set, partial-set, open-set, and open-partial-set) are provided in Table 1, 2, 3, 4, respectively. The results of optimizing only the textual prompts are provided in Appendix A.3.

**Results under closed-set category shift (Table 1).** Generally, UEO achieves competitive results under the closed-set category shift. When carefully comparing the results, we can find that InfoMax always obtains the best performance in terms of the ACC, and UEO always obtains the best or the second-best performance in terms of the AUC. Compared with InfoMax, the ACC scores of UEO are satisfying, indicating that it makes a good balance between ID generalization and OOD detection.

**Results under partial-set category shift (Table 2).** Our method achieves the highest accuracy on most tasks among the four benchmarks under partial-set category shift. For the AUC score, UEO also outperforms most baselines and nearly approaches the maximum of all methods. Notably, on two large-scale datasets, DomainNet and VISDA-C, UEO simultaneously obtains impressive results in both classifying ID categories and rejecting OOD categories.

Table 1: Results under the closed-set category shift (ResNet-50). [**Best** & second best]

| Methods | DN (Cl) | | DN (In) | | DN (Pa) | | DN (Qu) | | DN (Re) | | DN (Sk) | | **DN (Avg.)** | | **VD (Avg.)** | | **OH (Avg.)** | | **OF (Avg.)** | |
| Metrics (%) | ACC | AUC | ACC | AUC | ACC | AUC | ACC | AUC | ACC | AUC | ACC | AUC | ACC | AUC | ACC | AUC | ACC | AUC | ACC | AUC |
|---|---|---|---|---|---|---|---|---|---|---|---|---|---|---|---|---|---|---|---|---|
| CLIP (Radford et al., 2021) | 54.7 | 67.4 | 44.7 | 64.1 | 51.8 | 69.1 | 6.2 | **53.2** | 77.4 | 79.9 | 49.1 | 65.1 | 47.3 | 66.5 | 88.9 | 81.6 | 73.2 | 75.0 | 73.2 | 82.8 |
| UPL (Huang et al., 2022) | 57.7 | **68.4** | 49.8 | 67.0 | 54.1 | **70.5** | 12.0 | 51.7 | 79.6 | **80.5** | 52.9 | **66.7** | 51.0 | **67.5** | 89.0 | 81.7 | 76.0 | **76.5** | 77.0 | 83.0 |
| POUF (Tanwisuth et al., 2023) | **58.9** | 68.1 | 49.7 | 67.1 | 55.6 | 69.1 | 10.6 | 50.5 | 76.6 | 75.6 | 51.8 | 63.7 | 50.5 | 65.7 | 92.3 | 84.1 | 76.1 | 76.4 | 76.7 | 83.3 |
| DANCE (Saito et al., 2020) | 56.5 | 66.6 | 49.4 | **67.2** | 52.8 | 68.1 | 0.3 | 49.1 | 79.2 | 78.6 | 50.1 | 64.7 | 48.1 | 65.7 | 90.2 | **84.7** | 75.0 | 76.1 | 74.6 | 83.2 |
| EntMin (Wang et al., 2021) | 55.2 | 67.9 | 44.2 | 63.6 | 52.3 | 69.5 | 0.3 | 50.3 | 78.4 | 79.9 | 49.5 | 65.4 | 46.7 | 66.1 | 90.3 | 82.7 | 73.8 | 75.6 | 73.7 | 82.9 |
| InfoMax (Liang et al., 2020) | 58.8 | 68.3 | **50.2** | 66.6 | **56.5** | 68.0 | **12.4** | 49.9 | **79.8** | 79.4 | 54.1 | 65.5 | **52.0** | 66.3 | **92.6** | 81.4 | **77.0** | 75.8 | **79.2** | 82.5 |
| UEO | **58.9** | 68.3 | 50.7 | 67.0 | 56.5 | 70.4 | 12.4 | 52.7 | 78.8 | 78.8 | **54.2** | 65.9 | 51.9 | 67.2 | 92.2 | 84.6 | 76.8 | 75.6 | 78.1 | **84.1** |

Table 2: Results of different methods under the partial-set category shift (ResNet-50).

| Methods | DN (Cl) | | DN (In) | | DN (Pa) | | DN (Qu) | | DN (Re) | | DN (Sk) | | **DN (Avg.)** | | **VD (Avg.)** | | **OH (Avg.)** | | **OF (Avg.)** | |
| Metrics (%) | ACC | AUC | ACC | AUC | ACC | AUC | ACC | AUC | ACC | AUC | ACC | AUC | ACC | AUC | ACC | AUC | ACC | AUC | ACC | AUC |
|---|---|---|---|---|---|---|---|---|---|---|---|---|---|---|---|---|---|---|---|---|
| CLIP (Radford et al., 2021) | 54.7 | 67.4 | 44.7 | 64.1 | 51.8 | 69.1 | 6.2 | **53.2** | 77.4 | 79.9 | 49.1 | 65.1 | 47.3 | 66.5 | 88.9 | 81.6 | 73.2 | 75.0 | 73.2 | 82.8 |
| UPL (Huang et al., 2022) | 57.1 | 68.4 | 49.6 | 66.7 | 54.2 | **70.5** | 11.8 | 51.9 | 79.4 | **80.7** | 52.3 | **66.4** | 50.7 | 67.4 | 89.2 | 81.9 | 74.2 | 75.6 | 74.6 | 83.4 |
| POUF (Tanwisuth et al., 2023) | 57.6 | 68.0 | 49.1 | **66.9** | 55.4 | 69.0 | 10.3 | 51.1 | 76.5 | 76.2 | 53.1 | 65.1 | 50.3 | 66.1 | 89.8 | 80.9 | 75.8 | **76.4** | 75.2 | 83.2 |
| DANCE (Saito et al., 2020) | 57.7 | 66.7 | 48.9 | **66.9** | 53.0 | 68.3 | 0.3 | 49.1 | 79.1 | 78.5 | 51.9 | 65.3 | 48.5 | 65.8 | 87.6 | 79.6 | 74.3 | 75.5 | 73.6 | 82.7 |
| EntMin (Wang et al., 2021) | 55.2 | 67.8 | 44.4 | 63.7 | 52.2 | 69.5 | 0.3 | 50.3 | 78.4 | 79.9 | 49.6 | 65.3 | 46.7 | 66.1 | 89.7 | 81.8 | 73.7 | 75.4 | 73.5 | 82.8 |
| InfoMax (Liang et al., 2020) | 58.5 | **68.6** | 50.0 | 66.2 | 56.3 | 68.1 | 12.2 | 50.5 | 79.7 | 79.4 | 53.8 | 65.6 | 51.7 | 66.4 | 88.7 | 79.4 | 76.3 | 75.4 | 76.7 | 83.6 |
| UEO | 58.6 | 68.4 | 50.7 | 66.9 | 56.5 | 70.3 | 12.3 | 52.9 | 78.7 | 79.1 | 54.1 | 65.8 | 51.8 | 67.2 | 90.0 | 82.9 | 76.6 | 76.3 | 76.4 | 83.4 |

Table 3: Results of different methods under the open-set category shift (ResNet-50).

| Methods | DN (Cl) | | DN (In) | | DN (Pa) | | DN (Qu) | | DN (Re) | | DN (Sk) | | **DN (Avg.)** | | **VD (Avg.)** | | **OH (Avg.)** | | **OF (Avg.)** | |
| Metrics (%) | ACC | AUC | ACC | AUC | ACC | AUC | ACC | AUC | ACC | AUC | ACC | AUC | ACC | AUC | ACC | AUC | ACC | AUC | ACC | AUC |
|---|---|---|---|---|---|---|---|---|---|---|---|---|---|---|---|---|---|---|---|---|
| CLIP (Radford et al., 2021) | 54.7 | 67.4 | 44.7 | 64.1 | 51.8 | 69.1 | 6.2 | **53.2** | 77.4 | 79.9 | 49.1 | 65.1 | 47.3 | 66.5 | 88.9 | 81.6 | 73.2 | 75.0 | 73.2 | 82.8 |
| UPL (Huang et al., 2022) | 57.9 | 68.3 | **50.2** | 67.2 | 54.1 | **70.4** | 12.1 | 52.0 | 79.5 | **80.4** | 53.3 | 66.3 | 51.2 | **67.4** | 89.1 | 81.8 | 76.0 | **76.7** | 76.7 | 82.3 |
| POUF (Tanwisuth et al., 2023) | 58.4 | 67.3 | 49.5 | **67.5** | 55.2 | 67.5 | 10.4 | 50.2 | 76.1 | 75.6 | 50.8 | 62.5 | 50.1 | 65.1 | 91.8 | 78.7 | 75.8 | 75.4 | 77.0 | 82.8 |
| DANCE (Saito et al., 2020) | 57.0 | 67.1 | 49.8 | **67.5** | 52.6 | 67.6 | 0.3 | 50.1 | 78.9 | 77.7 | 44.5 | 60.4 | 47.2 | 65.1 | 61.4 | 55.2 | 74.5 | 75.6 | 74.5 | 82.7 |
| EntMin (Wang et al., 2021) | 55.3 | 67.7 | 43.6 | 63.3 | 52.1 | 69.3 | 0.3 | 50.2 | 78.5 | 79.8 | 49.4 | 65.4 | 46.5 | 66.0 | 89.8 | 81.6 | 73.8 | 75.4 | 73.7 | 82.8 |
| InfoMax (Liang et al., 2020) | **58.8** | 67.7 | **50.2** | 66.3 | **56.4** | 67.4 | **12.3** | 50.1 | **79.6** | 78.3 | **54.1** | 65.0 | **51.9** | 65.8 | **92.5** | 77.5 | **77.0** | 74.4 | **78.8** | 80.9 |
| UEO | **58.8** | 68.3 | 51.0 | 66.9 | 56.5 | 70.4 | 12.5 | 52.4 | 79.0 | 80.2 | 54.0 | 66.3 | **51.9** | **67.4** | **92.6** | 81.2 | 76.8 | 75.0 | 77.9 | 82.9 |
| UEO(O) | 55.4 | 71.3 | 50.0 | 69.9 | 53.2 | 75.8 | 9.3 | 58.2 | 76.0 | 87.8 | 51.0 | 71.9 | 49.1 | 72.5 | 93.4 | 85.4 | 74.2 | 82.6 | 75.0 | 88.3 |

**Results under open-set category shift (Table 3).** UEO demonstrates its effectiveness in scenarios where the training data contains OOD samples. Although UPL also performs well in rejecting OOD

Table 4: Results of different methods under the open-partial-set category shift (ResNet-50).

| Methods Metrics (%) | DN (Cl) ACC | AUC | DN (In) ACC | AUC | DN (Pa) ACC | AUC | DN (Qu) ACC | AUC | DN (Re) ACC | AUC | DN (Sk) ACC | AUC | DN (Avg.) ACC | AUC | VD (Avg.) ACC | AUC | OH (Avg.) ACC | AUC | OF (Avg.) ACC | AUC |
|---|---|---|---|---|---|---|---|---|---|---|---|---|---|---|---|---|---|---|---|---|
| CLIP (Radford et al., 2021) | 54.7 | 67.4 | 44.7 | 64.1 | 51.8 | 69.1 | 6.2 | 53.2 | 77.4 | 79.9 | 49.1 | 65.1 | 47.3 | 66.5 | 88.9 | 81.6 | 73.2 | 75.0 | 73.2 | 82.8 |
| UPL (Huang et al., 2022) | 57.3 | 68.4 | 49.7 | 67.0 | 54.4 | 70.7 | 11.8 | 51.9 | 79.5 | 80.6 | 53.2 | 66.5 | 51.0 | 67.5 | 89.8 | 82.5 | 74.9 | 75.8 | 75.3 | 82.5 |
| POUF (Tanwisuth et al., 2023) | 58.3 | 68.0 | 50.2 | 67.1 | 55.0 | 68.1 | 10.4 | 51.2 | 76.6 | 75.3 | 52.3 | 63.9 | 50.5 | 65.6 | 91.6 | 74.5 | 75.6 | 75.8 | 75.4 | 82.4 |
| DANCE (Saito et al., 2020) | 57.5 | 67.4 | 49.2 | 67.1 | 52.7 | 67.9 | 0.3 | 49.2 | 78.5 | 77.4 | 50.9 | 64.5 | 48.2 | 65.6 | 88.5 | 74.8 | 74.6 | 75.2 | 73.5 | 82.5 |
| EntMin (Wang et al., 2021) | 55.2 | 67.8 | 43.8 | 63.4 | 52.0 | 69.3 | 0.3 | 50.2 | 78.4 | 79.8 | 49.6 | 65.3 | 46.6 | 66.0 | 89.1 | 80.5 | 73.7 | 75.3 | 73.4 | 82.7 |
| InfoMax (Liang et al., 2020) | 58.7 | 68.2 | 50.0 | 65.8 | 56.6 | 68.1 | 12.3 | 50.3 | 79.5 | 78.1 | 54.0 | 65.3 | 51.9 | 66.0 | 90.6 | 74.6 | 76.5 | 73.9 | 77.2 | 81.5 |
| UEO | 58.7 | 68.2 | 50.8 | 66.8 | 56.4 | 70.7 | 12.7 | 52.7 | 78.7 | 80.2 | 53.9 | 66.1 | 51.9 | 67.4 | 92.0 | 81.2 | 76.5 | 75.6 | 77.2 | 83.2 |
| UEO(O) | 55.5 | 70.5 | 50.5 | 70.1 | 53.4 | 75.1 | 9.9 | 57.9 | 76.0 | 86.9 | 51.7 | 70.8 | 49.5 | 71.9 | 89.6 | 78.5 | 74.1 | 81.4 | 73.3 | 88.0 |

samples, it falls short in generalizing to ID categories. In contrast, UEO strikes a good balance between ID generalization and OOD detection. For instance, on the DomainNet dataset and the optimization of both text prompt and visual normalization layers, UEO achieves an accuracy of 51.9% and an AUC score of 67.4%.

**Results under open-partial-set category shift (Table 4).** As the most complicated shift in U$^2$-FT, the open-partial setting tests the ability of adaptation algorithms to handle both missing and OOD categories. UEO obtains better results compared to the baseline methods on this category shift. On the widely used Office dataset, UEO achieves an accuracy of 77.2% and an AUC score of 83.2%, which is the only method that outperforms zero-shot inference of CLIP.

## 4.3 ANALYSIS

**ViT-based experiments.** We further adopt the ViT-B/16 backbone (Radford et al., 2021) as the pre-trained model and report the results in Table 5. From the results, we observe that UEO achieves competitive performance on the DomainNet dataset under all four settings. In comparison to Info-Max and UPL, our method exhibits better ID generalization and OOD detection ability.

Table 5: Results of different methods on DomainNet (ViT-B/16).

| Methods Metrics (%) | closed-set ACC | AUC | partial-set ACC | AUC | open-set ACC | AUC | open-partial ACC | AUC | Avg. ACC | AUC |
|---|---|---|---|---|---|---|---|---|---|---|
| CLIP (Radford et al., 2021) | 58.2 | 72.6 | 58.2 | 72.6 | 58.2 | 72.6 | 58.2 | 72.6 | 58.2 | 72.6 |
| UPL (Huang et al., 2022) | 61.5 | 72.8 | 61.2 | 73.0 | 61.7 | 72.9 | 61.4 | 73.0 | 61.4 | 72.9 |
| POUF (Tanwisuth et al., 2023) | 60.9 | 71.1 | 61.4 | 71.6 | 61.0 | 70.8 | 61.1 | 70.8 | 61.1 | 71.1 |
| DANCE (Saito et al., 2020) | 57.9 | 72.0 | 57.9 | 72.0 | 57.8 | 71.6 | 57.9 | 71.7 | 57.9 | 71.7 |
| EntMin (Wang et al., 2021) | 56.0 | 71.6 | 56.0 | 71.7 | 55.7 | 71.3 | 55.8 | 71.4 | 55.9 | 71.5 |
| InfoMax (Liang et al., 2020) | 62.2 | 71.0 | 62.2 | 70.9 | 62.3 | 70.6 | 62.1 | 70.5 | 62.2 | 70.7 |
| UEO | 62.0 | 72.5 | 61.9 | 72.5 | 62.1 | 73.0 | 62.0 | 72.9 | 62.0 | 72.7 |

**Conventional closed-set setting.** Following the setting of POUF (Tanwisuth et al., 2023), we evaluate the effectiveness of UEO under a conventional closed-set setting, which considers all categories as ID. We adopt global-level accuracy as the experimental metric. From Table 6, UEO improves accuracy on DomainNet from 57.6% to 61.1% and outperforms all baseline methods. As InfoMax leverages the closed-set setting as prior knowledge, we treat its performance as the oracle results.

Table 6: Results of conventional closed-set setting on DomainNet (ResNet-50 and ViT-B/16).

| Backbones | ResNet-50 | | | | | | | ViT-B/16 | | | | | | |
|---|---|---|---|---|---|---|---|---|---|---|---|---|---|---|
| Methods | Cl | In | Pa | Qu | Re | Sk | Avg. | Cl | In | Pa | Qu | Re | Sk | Avg. |
| CLIP (Radford et al., 2021) | 54.8 | 40.9 | 54.6 | 6.0 | 77.7 | 49.2 | 47.2 | 71.0 | 47.6 | 66.2 | 13.9 | 83.7 | 63.5 | 57.6 |
| UPL (Huang et al., 2022) | 58.1 | 45.8 | 56.6 | 11.1 | 79.6 | 52.8 | 50.7 | 72.4 | 53.9 | 66.8 | 19.9 | 84.8 | 65.8 | 60.6 |
| POUF (Tanwisuth et al., 2023) | 58.9 | 45.8 | 58.1 | 9.6 | 76.6 | 50.7 | 50.0 | 72.3 | 53.3 | 69.8 | 18.0 | 83.3 | 65.3 | 60.3 |
| DANCE (Saito et al., 2020) | 57.2 | 44.7 | 57.1 | 0.3 | 79.1 | 10.9 | 41.6 | 72.9 | 52.7 | 67.8 | 0.3 | 85.3 | 66.0 | 57.5 |
| EntMin (Wang et al., 2021) | 55.6 | 39.9 | 55.9 | 0.3 | 78.7 | 49.8 | 46.7 | 71.3 | 45.9 | 67.3 | 0.3 | 84.3 | 63.4 | 55.4 |
| InfoMax (Liang et al., 2020) | 59.4 | 45.7 | 59.3 | 11.7 | 79.9 | 53.5 | 51.6 | 73.6 | 53.9 | 69.7 | 20.3 | 85.7 | 66.2 | 61.6 |
| UEO | 59.7 | 46.6 | 59.2 | 11.8 | 78.9 | 53.8 | 51.7 | 73.5 | 54.2 | 69.2 | 18.9 | 84.5 | 66.4 | 61.1 |

**Choices of optimization strategies.** We investigate the performance of UEO and the baseline method POUF under different optimization strategies. To validate the stability of the training process, we selected 6 combinations of textual prompt (P), BatchNorm (BN), and bias (BIAS) as the test range, according to (Zaken et al., 2022). The results depicted in Fig. 3 demonstrate that UEO is stable and outperforms POUF in various cases. The optimization of the vision encoder leads to better performance, indicating that UEO can benefit from a larger parameter space.

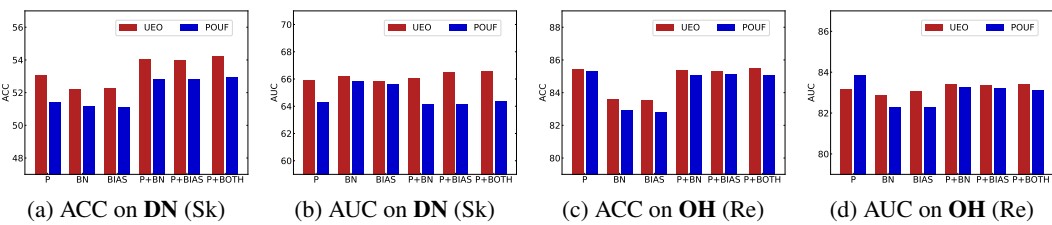

Figure 3: Results of different optimization designs on two tasks (**DN** (Sk) and **OH** (Re), ResNet-50).

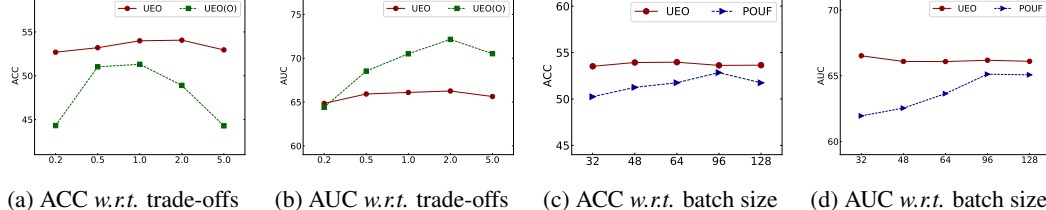

Figure 4: Results of different hyperparameters on task **DN** (Sk) (ResNet-50).

**Hyperparameter sensitivity.** Even UEO does not require hyperparameter tuning, we present the results of UEO under varying trade-offs (0.2, 0.5, 1.0, 2.0, 5.0) in the loss function and batch size (32, 48, 64, 96, 128) during training in Fig. 4. The results demonstrate that UEO remains stable across changes in the trade-off. Although UEO(O) can achieve better detection ability with a larger trade-off, its classification accuracy drops significantly. Additionally, UEO is not sensitive to changes in batch size. In contrast, POUF requires a larger batch size to estimate the distribution and performs poorly on smaller batches. Therefore, UEO does not suffer from hyperparameter selection.

**Different choices of $\Phi(\cdot)$ and prompt initialization.** Table 7 contains the results of different monotonically decreasing function designs of $\Phi(\cdot)$ in Eq. (4) and textual prompt initialization under the open-partial-set category shift. We find that UEO achieves nearly consistent performance under various function designs. For prompt initialization, the different designs obtain similar results with the choice ("a photo of a"), achieving better performance for both ID generalization and OOD detection.

Table 7: Results of different choices of $\Phi(\cdot)$ and prompt initialization on **DN** (Sk) (ResNet-50).

| (a) Weight function | | | (b) Prompt initialization | | | | |
|---|---|---|---|---|---|---|---|
| $\Phi(w)$ | ACC | AUC | Prompt initialization | ACC (CLIP) | AUC (CLIP) | ACC (UEO) | AUC (UEO) |
| $1/w$ | 53.9 | 66.1 | 'a photo of a' | 49.1 | 65.1 | 54.0 | 66.1 |
| $\sqrt{1/w}$ | 54.1 | 66.4 | 'a photo of many' | 47.9 | 65.0 | 52.9 | 66.3 |
| $(1/w)^2$ | 53.7 | 65.9 | 'a sketch of a' | 52.2 | 65.4 | 54.5 | 65.9 |
| $1-w$ | 53.9 | 66.4 | 'a painting of a' | 50.3 | 65.5 | 54.0 | 65.7 |
| $\sqrt{1-w}$ | 54.2 | 66.3 | 'this is a photo of a' | 49.3 | 64.9 | 53.2 | 66.1 |
| $(1-w)^2$ | 53.9 | 66.2 | 'this is a picture of a' | 49.2 | 67.4 | 54.3 | 65.9 |

## 5 CONCLUSION

In this paper, we introduce a novel unsupervised universal fine-tuning ($U^2$-FT) setting for VLMs, which does not rely on prior knowledge about the unlabeled data in the downstream domain. In addition to assessing the generalization ability to identify samples from the candidate classes, $U^2$-FT also considers improving the detection of OOD samples outside the list of candidate classes after fine-tuning. We propose Universal Entropy Optimization (UEO), which employs sample-level confidence to approximately minimize the entropy of ID samples and maximize the entropy of OOD samples. UEO is a simple and parameter-efficient approach that updates only a small number of parameters and does not require any hyper-parameters in the objective. Extensive results show that UEO always outperforms existing unsupervised fine-tuning methods across various category shift scenarios. We believe that the introduced $U^2$-FT setting is an interesting and important contribution to the field of transfer learning with VLMs and has the potential to attract significant attention. **Limitation.** While we have successfully validated the effectiveness of UEO for image classification, we have not yet studied its application to dense prediction tasks such as segmentation and detection.

## REPRODUCIBILITY STATEMENT

We have included the source code for reproducibility in the supplementary materials.

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

# A APPENDIX

## A.1 PSEUDO CODE

To facilitate a better understanding of our problem setup and proposed method (UEO), we provide the pseudocode below. Throughout this paper, we utilize two different scores, namely **ACC** and **AUC**, calculated in Line 15 and Line 17, respectively, to evaluate all methods under the $U^2$-FT framework.

---

**Algorithm 1** Universal Entropy Optimization (UEO) for Unsupervised Universal Fine-Tuning (U$^2$-FT)

---

1: # The training stage
2: **Input:** The pre-trained CLIP, unlabeled data $\mathcal{X}_t$ associated with the label set $L_u$, the name list of interested classes $L_p = \{y_1, \ldots, y_C\}$.
3: **for** $epoch = 1, 2, \ldots$ **do**
4:     **for** $iteration = 1, 2, \ldots$ **do**
5:         Sample a mini-batch $\mathcal{B}_t$ from $\mathcal{X}_t$.
6:         Forward $\mathcal{B}_t$ to the **original** CLIP to obtain the weights $w(x) = \mathcal{S}(x)$ using Eq. (2).
7:         Forward $\mathcal{B}_t$ to CLIP to obtain the predictions $\{p(x)\}$ using Eq. (1).
8:         Update the parameters of CLIP through the gradient of $\mathcal{L}$ in Eq. (4).
9:     **end for**
10: **end for**
11: # The testing stage
12: **Input:** The adapted CLIP, evaluation data $\mathcal{X}_e$ associated with the label set $L_e$, the name list of interested classes $L_p = \{y_1, \ldots, y_C\}$.
13: Split $\mathcal{X}_e$ into two sets, $\mathcal{X}_e^1$ associated with $L_e \cap L_p$ and $\mathcal{X}_e^2$ associated with $L_e \setminus L_p$.
14: Forward $\mathcal{X}_e$ to the adapted CLIP to obtain the predictions $\{p(x)\}$ using Eq. (1).
15: Calculate the per-class accuracy (**ACC**) over $\mathcal{X}_e^1$ based on the argmax operation.
16: Obtain the scores $\mathcal{S}(x)$ for both $\mathcal{X}_e^1$ and $\mathcal{X}_e^2$ using Eq. (2).
17: Calculate the **AUC** score to measure the ability of outlier detection.

---

## A.2   INFORMATION OF DATA SPLIT FOR DIFFERENT SHIFTS

In this section, we present specific information about $L_p$ (the target class of interest), $L_u$ (the label set of the training data), and $L_e$ (the label set of the evaluation data) for all datasets.

Table 8: Detailed information about four category shifts in the training stage and evaluation stage.

| Datasets | Category Shifts | $L_p$ | $L_e$ | $L_u$ |
|---|---|---|---|---|
| DomainNet (DN) | closed-set | $[0, 300)$ | $[0, 345)$ | $[0, 300)$ |
| | partial-set | $[0, 300)$ | $[0, 345)$ | $[0, 250)$ |
| | open-set | $[0, 300)$ | $[0, 345)$ | $[0, 330)$ |
| | open-partial-set | $[0, 300)$ | $[0, 345)$ | $[0, 250) \cup [300, 330)$ |
| VISDA-C (VD) | closed-set | $[0, 8)$ | $[0, 12)$ | $[0, 8)$ |
| | partial-set | $[0, 8)$ | $[0, 12)$ | $[0, 6)$ |
| | open-set | $[0, 8)$ | $[0, 12)$ | $[0, 10)$ |
| | open-partial-set | $[0, 8)$ | $[0, 12)$ | $[0, 6) \cup [8, 10)$ |
| OfficeHome (OH) | closed-set | $[0, 50)$ | $[0, 65)$ | $[0, 50)$ |
| | partial-set | $[0, 50)$ | $[0, 65)$ | $[0, 35)$ |
| | open-set | $[0, 50)$ | $[0, 65)$ | $[0, 60)$ |
| | open-partial-set | $[0, 50)$ | $[0, 65)$ | $[0, 35) \cup [50, 60)$ |
| Office (OF) | closed-set | $[0, 25)$ | $[0, 31)$ | $[0, 25)$ |
| | partial-set | $[0, 25)$ | $[0, 31)$ | $[0, 15)$ |
| | open-set | $[0, 25)$ | $[0, 31)$ | $[0, 28)$ |
| | open-partial-set | $[0, 25)$ | $[0, 31)$ | $[0, 15) \cup [25, 28)$ |

## A.3   ADDITIONAL RESULTS OF DIFFERENT METHODS UNDER PROMPT TUNING

As for only prompt tuning on the textural branch, the results of different methods under four distinct category shift scenarios (*i.e.*, closed-set, partial-set, open-set, and open-partial-set) are provided in Table 9, 10, 11, 12, respectively.

Table 9: Results of different methods under the closed-set category shift (ResNet-50).

| Methods | DN (Cl) | | DN (In) | | DN (Pa) | | DN (Qu) | | DN (Re) | | DN (Sk) | | DN (Avg.) | | VD (Avg.) | | OH (Avg.) | | OF (Avg.) | |
|---|---|---|---|---|---|---|---|---|---|---|---|---|---|---|---|---|---|---|---|---|
| Metrics (%) | ACC | AUC | ACC | AUC | ACC | AUC | ACC | AUC | ACC | AUC | ACC | AUC | ACC | AUC | ACC | AUC | ACC | AUC | ACC | AUC |
| CLIP (Radford et al., 2021) | 54.7 | 67.4 | 44.7 | 64.1 | 51.8 | 69.1 | 6.2 | 53.2 | 77.4 | 79.9 | 49.1 | 65.1 | 47.3 | 66.5 | 88.9 | 81.6 | 73.2 | 75.0 | 73.2 | 82.8 |
| UPL (Huang et al., 2022) | 56.8 | 67.9 | 49.3 | 66.8 | 53.2 | 70.6 | 9.7 | 52.2 | 79.5 | 80.4 | 52.1 | 65.8 | 50.1 | 67.3 | 89.0 | 81.9 | 75.6 | 75.7 | 76.8 | 82.4 |
| POUF (Tanwisuth et al., 2023) | 58.4 | 67.9 | 49.6 | 67.0 | 55.1 | 68.8 | 8.1 | 53.5 | 76.8 | 76.0 | 51.7 | 64.4 | 49.9 | 66.3 | 92.1 | 83.9 | 75.4 | 76.3 | 76.2 | 83.0 |
| DANCE (Saito et al., 2020) | 57.6 | 67.2 | 49.6 | 67.4 | 51.2 | 66.1 | 0.4 | 49.8 | 78.6 | 78.7 | 45.9 | 63.0 | 47.2 | 65.4 | 92.5 | 82.1 | 74.5 | 75.9 | 74.8 | 83.2 |
| EntMin (Wang et al., 2021) | 54.9 | 67.4 | 44.0 | 63.7 | 51.9 | 69.7 | 0.3 | 49.5 | 78.3 | 80.0 | 48.9 | 65.2 | 46.4 | 65.9 | 89.0 | 81.9 | 73.2 | 75.1 | 73.3 | 82.8 |
| InfoMax (Liang et al., 2020) | 57.9 | 68.0 | 49.3 | 66.4 | 55.1 | 68.4 | 9.0 | 54.1 | 79.4 | 79.4 | 52.9 | 65.5 | 50.6 | 67.0 | 91.8 | 81.2 | 76.0 | 75.7 | 78.6 | 82.3 |
| UEO | 58.4 | 67.9 | 50.1 | 66.8 | 56.3 | 71.0 | 8.3 | 53.6 | 78.7 | 78.9 | 53.1 | 65.9 | 50.8 | 67.3 | 91.1 | 84.4 | 76.2 | 74.9 | 77.6 | 83.5 |

Table 10: Results of different methods under the partial-set category shift (ResNet-50).

| Methods | DN (Cl) | | DN (In) | | DN (Pa) | | DN (Qu) | | DN (Re) | | DN (Sk) | | DN (Avg.) | | VD (Avg.) | | OH (Avg.) | | OF (Avg.) | |
|---|---|---|---|---|---|---|---|---|---|---|---|---|---|---|---|---|---|---|---|---|
| Metrics (%) | ACC | AUC | ACC | AUC | ACC | AUC | ACC | AUC | ACC | AUC | ACC | AUC | ACC | AUC | ACC | AUC | ACC | AUC | ACC | AUC |
| CLIP (Radford et al., 2021) | 54.7 | 67.4 | 44.7 | 64.1 | 51.8 | 69.1 | 6.2 | 53.2 | 77.4 | 79.9 | 49.1 | 65.1 | 47.3 | 66.5 | 88.9 | 81.6 | 73.2 | 75.0 | 73.2 | 82.8 |
| UPL (Huang et al., 2022) | 56.2 | 67.8 | 49.0 | 66.5 | 53.1 | 70.7 | 9.5 | 52.2 | 79.5 | 80.8 | 51.0 | 65.9 | 49.7 | 67.3 | 89.0 | 81.7 | 73.7 | 74.9 | 74.3 | 83.1 |
| POUF (Tanwisuth et al., 2023) | 57.1 | 67.8 | 49.0 | 67.1 | 55.4 | 69.5 | 8.0 | 53.8 | 77.3 | 75.8 | 52.4 | 64.8 | 49.9 | 66.5 | 88.3 | 78.3 | 74.8 | 76.1 | 75.1 | 83.0 |
| DANCE (Saito et al., 2020) | 57.9 | 66.9 | 49.7 | 67.3 | 51.8 | 67.5 | 0.3 | 49.9 | 79.1 | 78.8 | 51.5 | 65.2 | 48.4 | 65.9 | 86.7 | 74.1 | 74.4 | 76.1 | 74.0 | 82.8 |
| EntMin (Wang et al., 2021) | 55.0 | 67.5 | 44.1 | 63.8 | 51.8 | 69.6 | 0.3 | 49.3 | 78.3 | 79.9 | 49.1 | 65.1 | 46.4 | 65.9 | 89.0 | 81.8 | 73.1 | 75.1 | 73.3 | 82.8 |
| InfoMax (Liang et al., 2020) | 57.7 | 68.3 | 49.1 | 66.1 | 55.6 | 68.6 | 8.9 | 54.1 | 79.4 | 79.3 | 52.8 | 65.5 | 50.6 | 67.0 | 87.8 | 79.8 | 75.3 | 75.5 | 75.6 | 83.2 |
| UEO | 58.1 | 67.9 | 50.2 | 66.7 | 56.2 | 70.9 | 8.5 | 53.4 | 78.6 | 79.0 | 53.2 | 66.0 | 50.8 | 67.3 | 89.0 | 82.2 | 75.9 | 75.9 | 75.8 | 82.9 |

Table 11: Results of different methods under the open-set category shift (ResNet-50).

| Methods | DN (Cl) | | DN (In) | | DN (Pa) | | DN (Qu) | | DN (Re) | | DN (Sk) | | DN (Avg.) | | VD (Avg.) | | OH (Avg.) | | OF (Avg.) | |
|---|---|---|---|---|---|---|---|---|---|---|---|---|---|---|---|---|---|---|---|---|
| Metrics (%) | ACC | AUC | ACC | AUC | ACC | AUC | ACC | AUC | ACC | AUC | ACC | AUC | ACC | AUC | ACC | AUC | ACC | AUC | ACC | AUC |
| CLIP (Radford et al., 2021) | 54.7 | 67.4 | 44.7 | 64.1 | 51.8 | 69.1 | 6.2 | 53.2 | 77.4 | 79.9 | 49.1 | 65.1 | 47.3 | 66.5 | 88.9 | 81.6 | 73.1 | 75.0 | 73.3 | 82.8 |
| UPL (Huang et al., 2022) | 56.9 | 68.0 | 49.8 | 67.1 | 53.3 | 70.6 | 9.7 | 52.3 | 79.5 | 80.3 | 52.5 | 66.1 | 50.3 | 67.4 | 89.0 | 81.8 | 75.8 | 75.9 | 76.5 | 81.9 |
| POUF (Tanwisuth et al., 2023) | 57.7 | 67.0 | 49.9 | 66.7 | 54.8 | 68.5 | 7.8 | 53.5 | 77.3 | 76.4 | 51.7 | 64.3 | 49.9 | 66.1 | 91.0 | 78.4 | 74.9 | 75.5 | 76.5 | 82.6 |
| DANCE (Saito et al., 2020) | 56.8 | 67.0 | 49.3 | 67.7 | 49.9 | 64.6 | 0.3 | 50.4 | 78.0 | 76.7 | 7.5 | 41.8 | 40.3 | 61.4 | 89.5 | 75.6 | 73.6 | 75.5 | 74.8 | 82.5 |
| EntMin (Wang et al., 2021) | 55.0 | 67.4 | 43.6 | 63.5 | 51.8 | 69.6 | 0.3 | 49.4 | 78.4 | 80.0 | 48.8 | 65.2 | 46.3 | 65.8 | 88.9 | 81.6 | 73.2 | 75.1 | 73.3 | 82.8 |
| InfoMax (Liang et al., 2020) | 57.6 | 67.4 | 49.0 | 65.8 | 55.2 | 67.5 | 9.1 | 53.7 | 79.3 | 78.4 | 52.9 | 65.1 | 50.5 | 66.3 | 91.2 | 73.8 | 75.6 | 74.6 | 78.3 | 80.3 |
| UEO | 58.3 | 68.1 | 50.1 | 66.7 | 56.2 | 70.9 | 8.5 | 53.8 | 78.9 | 80.0 | 52.9 | 66.3 | 50.8 | 67.6 | 91.3 | 79.2 | 76.0 | 74.7 | 78.1 | 82.7 |
| UEO(O) | 55.4 | 71.8 | 49.2 | 69.7 | 53.0 | 75.1 | 6.7 | 59.8 | 75.6 | 86.8 | 49.9 | 71.3 | 48.3 | 72.4 | 92.9 | 83.3 | 72.1 | 81.8 | 75.3 | 88.7 |

Table 12: Results of different methods under the open-partial-set category shift (ResNet-50).

| Methods | DN (Cl) | | DN (In) | | DN (Pa) | | DN (Qu) | | DN (Re) | | DN (Sk) | | DN (Avg.) | | VD (Avg.) | | OH (Avg.) | | OF (Avg.) | |
|---|---|---|---|---|---|---|---|---|---|---|---|---|---|---|---|---|---|---|---|---|
| Metrics (%) | ACC | AUC | ACC | AUC | ACC | AUC | ACC | AUC | ACC | AUC | ACC | AUC | ACC | AUC | ACC | AUC | ACC | AUC | ACC | AUC |
| CLIP (Radford et al., 2021) | 54.7 | 67.4 | 44.7 | 64.1 | 51.8 | 69.1 | 6.2 | 53.2 | 77.4 | 79.9 | 49.1 | 65.1 | 47.3 | 66.5 | 88.9 | 81.6 | 73.2 | 75.0 | 73.2 | 82.8 |
| UPL (Huang et al., 2022) | 56.5 | 67.8 | 49.2 | 66.8 | 53.5 | 70.8 | 9.6 | 51.8 | 79.5 | 80.8 | 52.0 | 65.8 | 50.0 | 67.3 | 89.3 | 82.0 | 74.7 | 75.1 | 74.7 | 82.5 |
| POUF (Tanwisuth et al., 2023) | 58.1 | 67.7 | 49.1 | 66.4 | 55.0 | 69.0 | 8.2 | 53.5 | 77.1 | 75.5 | 51.4 | 64.3 | 49.8 | 66.1 | 90.6 | 72.2 | 74.9 | 76.0 | 75.3 | 82.5 |
| DANCE (Saito et al., 2020) | 57.8 | 66.4 | 49.3 | 67.5 | 51.7 | 65.6 | 0.3 | 49.7 | 78.2 | 77.2 | 50.2 | 64.0 | 47.9 | 65.2 | 85.8 | 74.1 | 74.5 | 75.2 | 74.2 | 82.9 |
| EntMin (Wang et al., 2021) | 54.9 | 67.4 | 43.7 | 63.6 | 51.7 | 69.6 | 0.3 | 49.3 | 78.3 | 79.9 | 49.1 | 65.1 | 46.3 | 65.8 | 88.9 | 81.4 | 73.1 | 75.1 | 73.2 | 82.7 |
| InfoMax (Liang et al., 2020) | 57.6 | 67.9 | 49.0 | 65.6 | 55.5 | 68.1 | 9.0 | 54.1 | 79.4 | 78.1 | 53.0 | 64.8 | 50.6 | 66.4 | 84.4 | 67.6 | 75.4 | 74.5 | 76.3 | 81.6 |
| UEO | 57.7 | 68.0 | 50.0 | 66.6 | 56.2 | 70.7 | 8.5 | 53.9 | 78.7 | 80.0 | 53.1 | 66.0 | 50.7 | 67.6 | 87.5 | 74.7 | 75.6 | 75.4 | 76.4 | 83.2 |
| UEO(O) | 56.2 | 70.4 | 49.8 | 69.5 | 52.3 | 74.7 | 6.7 | 59.4 | 76.1 | 86.7 | 50.8 | 69.7 | 48.7 | 71.7 | 84.8 | 77.1 | 72.9 | 80.4 | 72.6 | 87.7 |

### A.4 ADDITIONAL ANALYSIS ON ANOTHER CATEGORY SHIFT

In addition to the primary task of **DN** (Sk) analyzed under the open-partial-set category shift in the main text, we also present the results of **DN** (Sk) under the closed-set one in Fig. 5 and Fig. 6. Furthermore, the closed-set shift, we list the results of different choices of $\Phi(\cdot)$ and prompt initialization in Table 13.

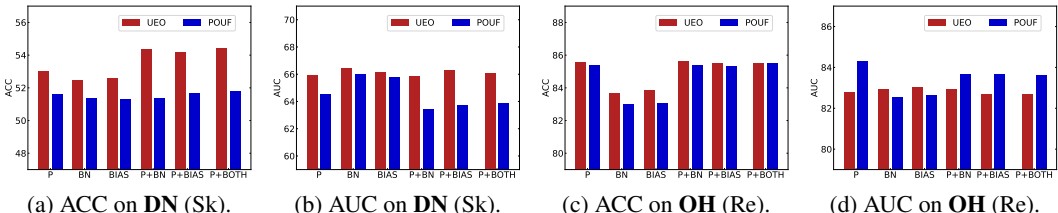

| (a) ACC on **DN** (Sk). | (b) AUC on **DN** (Sk). | (c) ACC on **OH** (Re). | (d) AUC on **OH** (Re). |
|---|---|---|---|

Figure 5: Results of different optimization designs on two tasks (**DN** (Sk) and **OH** (Re)) (ResNet-50).

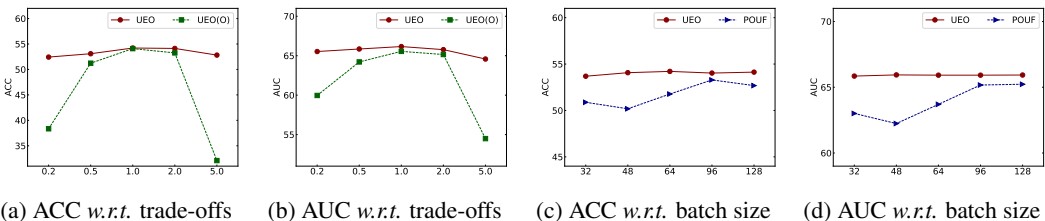

| (a) ACC *w.r.t.* trade-offs | (b) AUC *w.r.t.* trade-offs | (c) ACC *w.r.t.* batch size | (d) AUC *w.r.t.* batch size |
|---|---|---|---|

Figure 6: Results of different hyperparameters on task **DN** (Sk) (ResNet-50, closed-set).

Table 13: Results of different choices of $\Phi(\cdot)$ and prompt initialization on **DN** (Sk) (closed-set).

(a) Weight function

| $\Phi(w)$ | ACC | AUC |
|---|---|---|
| $1/w$ | 54.3 | 65.9 |
| $\sqrt{1/w}$ | 54.2 | 66.0 |
| $(1/w)^2$ | 53.6 | 65.7 |
| $1-w$ | 54.2 | 65.9 |
| $\sqrt{1-w}$ | 54.1 | 66.0 |
| $(1-w)^2$ | 54.1 | 66.0 |

(b) Prompt initialization

| Prompt initialization | ACC (CLIP) | AUC (CLIP) | ACC(UEO) | AUC (UEO) |
|---|---|---|---|---|
| 'a photo of a' | 49.1 | 65.1 | 54.4 | 65.9 |
| 'a photo of many' | 47.9 | 65.0 | 53.1 | 66.1 |
| 'a sketch of a' | 52.2 | 65.4 | 54.6 | 65.8 |
| 'a painting of a' | 50.3 | 65.5 | 53.9 | 66.0 |
| 'this is a photo of a' | 49.3 | 64.9 | 53.0 | 65.8 |
| 'this is a picture of a' | 49.2 | 64.6 | 54.2 | 65.9 |

### A.5 DETAILED RESULTS OF DIFFERENT DATASETS (VD, OH, OF) IN THE MAIN TEXT

Rather than presenting the averaged results shown in Table 1 ∼ Table 4, we present the complete results of all methods on three datasets (VD, OH, OF) in Table 14 ∼ Table 17.

Table 14: Results of different methods under the closed-set category shift (ResNet-50).

| Methods | VD (T) | | VD (V) | | OH (Ar) | | OH (Cl) | | OH (Pr) | | OH (Re) | | OF (A) | | OF (D) | | OF (W) | |
|---|---|---|---|---|---|---|---|---|---|---|---|---|---|---|---|---|---|---|
| Metrics (%) | ACC | AUC | ACC | AUC | ACC | AUC | ACC | AUC | ACC | AUC | ACC | AUC | ACC | AUC | ACC | AUC | ACC | AUC |
| CLIP (Radford et al., 2021) | 88.2 | 81.2 | 89.5 | 82.1 | 70.2 | 74.9 | 55.3 | 64.7 | 84.2 | 78.0 | 83.1 | 82.5 | 76.6 | 84.8 | 73.6 | 85.6 | 69.5 | 77.9 |
| UPL (Huang et al., 2022) | 88.2 | 81.2 | 89.8 | 82.5 | 71.9 | 74.9 | 58.4 | 63.9 | 86.3 | 79.8 | 85.6 | 84.2 | 80.1 | 79.7 | 76.7 | **87.8** | 73.7 | 79.6 |
| POUF (Tanwisuth et al., 2023) | 93.1 | 83.4 | **91.0** | 84.3 | 71.2 | **75.4** | 59.1 | 64.8 | 85.9 | 80.5 | 85.4 | **84.3** | 78.9 | 82.0 | 75.1 | 87.0 | 74.6 | 79.8 |
| DANCE (Saito et al., 2020) | **94.7** | 80.5 | 90.2 | 83.8 | 70.5 | 74.9 | 57.1 | 63.8 | 86.0 | **81.5** | 84.3 | 83.6 | 79.8 | **85.5** | 74.0 | 85.9 | 70.6 | 78.3 |
| EntMin (Wang et al., 2021) | 88.1 | 81.0 | 90.0 | 82.7 | 70.5 | 74.8 | 54.6 | 65.2 | 84.5 | 78.0 | 83.3 | 82.5 | 76.6 | 84.8 | 73.5 | 85.6 | 69.7 | 77.9 |
| InfoMax (Liang et al., 2020) | 94.5 | 83.2 | 89.1 | 79.3 | 72.9 | 73.3 | 57.7 | **65.6** | **87.7** | 80.9 | **85.9** | 82.9 | **80.4** | 79.3 | **76.9** | 87.4 | **78.4** | 80.3 |
| UEO | 91.5 | **84.0** | 90.8 | **84.8** | **73.1** | 74.0 | **59.5** | 64.5 | 86.8 | 78.3 | 85.4 | 82.8 | 80.2 | 82.9 | 75.4 | 87.0 | 77.1 | **80.7** |
| **Prompt Only ⇈, Prompt + Normalization Layers ⇊** | | | | | | | | | | | | | | | | | | |
| CLIP (Radford et al., 2021) | 88.2 | 81.2 | 89.5 | 82.1 | 70.1 | 74.9 | 55.3 | 64.7 | 84.2 | 78.0 | 83.1 | 82.5 | 76.6 | 84.8 | 73.6 | 85.6 | 69.5 | 77.9 |
| UPL (Huang et al., 2022) | 88.3 | 81.1 | 89.8 | 82.4 | 71.6 | 75.6 | 60.1 | 65.9 | 86.5 | 79.9 | 85.7 | 84.5 | 80.4 | 79.5 | 76.5 | **88.9** | 74.0 | 80.6 |
| POUF (Tanwisuth et al., 2023) | 93.9 | 82.6 | **90.8** | **85.6** | 71.7 | **75.9** | **61.6** | **66.7** | 85.7 | 79.5 | 85.4 | **83.6** | 79.7 | 82.4 | 75.6 | 87.4 | 74.6 | 80.2 |
| DANCE (Saito et al., 2020) | 90.5 | **84.8** | 89.8 | 84.5 | 70.6 | 75.1 | 59.9 | 65.5 | 85.1 | 80.4 | 84.5 | 83.4 | 79.7 | **85.7** | 73.6 | 85.8 | 70.6 | 78.1 |
| EntMin (Wang et al., 2021) | 90.4 | 82.4 | 90.2 | 83.0 | 70.6 | 75.1 | 56.3 | 65.8 | 85.0 | 78.8 | 83.4 | 82.8 | 77.6 | 84.9 | 73.7 | 85.8 | 69.7 | 78.0 |
| InfoMax (Liang et al., 2020) | **96.0** | 84.5 | 89.2 | 78.2 | **73.2** | 73.5 | 61.0 | 65.7 | **87.9** | **80.9** | **85.9** | 82.9 | **80.5** | 78.6 | **76.8** | 87.4 | **80.5** | 81.5 |
| UEO | 93.3 | 84.7 | **91.1** | 84.5 | 72.3 | 75.0 | 61.4 | 64.9 | 87.7 | 79.6 | 85.7 | 83.0 | 80.2 | 83.2 | 75.9 | 87.4 | 78.2 | **81.7** |

Table 15: Results of different methods under the partial-set category shift (ResNet-50).

| Methods / Metrics (%) | VD (T) ACC | AUC | VD (V) ACC | AUC | OH (Ar) ACC | AUC | OH (Cl) ACC | AUC | OH (Pr) ACC | AUC | OH (Re) ACC | AUC | OF (A) ACC | AUC | OF (D) ACC | AUC | OF (W) ACC | AUC |
|---|---|---|---|---|---|---|---|---|---|---|---|---|---|---|---|---|---|---|
| CLIP (Radford et al., 2021) | **88.2** | **81.2** | 89.5 | 82.1 | 70.2 | 74.9 | 55.3 | 64.7 | 84.2 | 78.0 | 83.1 | 82.5 | 76.6 | 84.8 | 73.6 | 85.6 | 69.5 | 77.9 |
| UPL (Huang et al., 2022) | 88.0 | 80.4 | 89.9 | 83.0 | 70.1 | 74.9 | 56.3 | 63.6 | 84.9 | 78.5 | 83.6 | 82.6 | 77.6 | 85.1 | 73.7 | 85.7 | 71.5 | 78.5 |
| POUF (Tanwisuth et al., 2023) | 86.7 | 73.1 | 90.0 | 83.5 | 71.1 | 75.2 | 57.8 | 65.2 | 85.7 | 80.4 | 84.6 | 83.7 | 79.5 | 84.1 | 74.2 | 86.1 | 71.7 | 78.6 |
| DANCE (Saito et al., 2020) | 84.9 | 67.1 | 88.5 | 81.1 | 70.6 | 74.9 | 57.4 | 65.1 | 85.4 | **81.2** | 84.0 | 83.3 | 78.3 | 84.8 | 73.8 | 85.7 | 70.0 | 78.0 |
| EntMin (Wang et al., 2021) | **88.2** | **81.2** | 89.8 | 82.3 | 70.2 | 74.8 | 54.8 | 65.1 | 84.3 | 78.0 | 83.1 | 82.5 | 76.6 | 84.8 | 73.6 | 85.6 | 69.6 | 77.9 |
| InfoMax (Liang et al., 2020) | 85.2 | 74.7 | 90.4 | **84.9** | 71.6 | 74.0 | 57.4 | **65.6** | 87.0 | 79.2 | 85.2 | 83.2 | **79.5** | 84.4 | 74.6 | **86.1** | 72.6 | **79.0** |
| UEO | 87.4 | 79.5 | **90.6** | **84.9** | **72.1** | 75.1 | **59.3** | 65.0 | 86.3 | 79.8 | **85.9** | **83.8** | **79.5** | 83.7 | **75.0** | **86.1** | **72.9** | 78.9 |
| *Prompt Only ⇈, Prompt + Normalization Layers⇊* | | | | | | | | | | | | | | | | | | |
| CLIP (Radford et al., 2021) | 88.2 | 81.2 | 89.5 | 82.1 | 70.1 | 74.9 | 55.3 | 64.7 | 84.2 | 78.0 | 83.1 | 82.5 | 76.6 | 84.8 | 73.6 | 85.6 | 69.5 | 77.9 |
| UPL (Huang et al., 2022) | 88.4 | 81.1 | 90.0 | 82.8 | 70.1 | 75.4 | 58.1 | 65.6 | 84.7 | 78.6 | 83.8 | 82.9 | 77.8 | 85.0 | 74.3 | 86.1 | 71.7 | 79.1 |
| POUF (Tanwisuth et al., 2023) | 89.9 | 77.2 | 89.8 | 84.6 | 72.0 | 75.6 | 60.7 | 66.1 | 85.6 | 80.2 | 84.8 | 83.5 | 79.3 | 84.5 | 74.7 | 86.0 | 71.8 | 79.0 |
| DANCE (Saito et al., 2020) | 85.4 | 75.1 | 89.8 | 84.0 | 70.5 | 75.0 | 58.1 | 65.5 | 84.6 | 78.5 | 84.2 | 83.1 | 77.4 | 84.7 | 73.8 | 85.7 | 69.6 | 77.8 |
| EntMin (Wang et al., 2021) | 89.4 | 81.2 | 90.0 | 82.4 | 70.4 | 75.1 | 56.1 | 65.5 | 85.1 | 78.5 | 83.2 | 82.7 | 77.1 | 84.8 | 73.8 | 85.7 | 69.5 | 78.0 |
| InfoMax (Liang et al., 2020) | 87.6 | 74.2 | 89.7 | 84.6 | 72.2 | 74.7 | 60.4 | 64.4 | 87.4 | 79.3 | 85.2 | 83.4 | 79.9 | 84.7 | 76.0 | 86.4 | 74.1 | 79.7 |
| UEO | 89.8 | 81.3 | 90.2 | 84.6 | 72.4 | 75.7 | 61.6 | 65.1 | 87.1 | 80.3 | 85.5 | 83.9 | 79.9 | 84.2 | 75.5 | 86.5 | 73.6 | 79.4 |

Table 16: Results of different methods under the open-set category shift (ResNet-50).

| Methods / Metrics (%) | VD (T) ACC | AUC | VD (V) ACC | AUC | OH (Ar) ACC | AUC | OH (Cl) ACC | AUC | OH (Pr) ACC | AUC | OH (Re) ACC | AUC | OF (A) ACC | AUC | OF (D) ACC | AUC | OF (W) ACC | AUC |
|---|---|---|---|---|---|---|---|---|---|---|---|---|---|---|---|---|---|---|
| CLIP (Radford et al., 2021) | 88.2 | 81.2 | 89.6 | 82.1 | 70.1 | 74.9 | 55.2 | 64.7 | 84.2 | 78.0 | 83.1 | 82.5 | 76.6 | 84.8 | 73.6 | 85.6 | 69.6 | 77.9 |
| UPL (Huang et al., 2022) | 88.2 | 81.2 | 89.8 | 82.5 | 72.1 | 75.1 | 59.1 | 64.6 | 86.4 | 79.8 | 85.6 | 84.3 | 80.2 | 79.4 | 75.8 | 87.3 | 73.6 | 78.9 |
| POUF (Tanwisuth et al., 2023) | 91.7 | 74.1 | 90.2 | 82.7 | 71.4 | 75.3 | 57.8 | 63.8 | 85.4 | 79.5 | 85.0 | 83.5 | 79.2 | 81.1 | 75.1 | 86.7 | 75.2 | 79.9 |
| DANCE (Saito et al., 2020) | 89.9 | 72.5 | 89.1 | 78.6 | 70.5 | 74.8 | 53.5 | 63.6 | 85.4 | 79.7 | 85.0 | 83.9 | 79.5 | 83.3 | 73.8 | 85.8 | 71.0 | 78.3 |
| EntMin (Wang et al., 2021) | 87.8 | 80.3 | 90.1 | 74.2 | 70.4 | 74.8 | 54.7 | 65.1 | 84.6 | 78.0 | 83.4 | 82.5 | 76.6 | 84.8 | 73.6 | 85.6 | 69.8 | 77.9 |
| InfoMax (Liang et al., 2020) | 94.0 | 70.5 | 88.3 | 77.1 | 71.9 | 72.0 | 57.5 | 64.9 | 87.4 | 79.5 | 85.7 | 82.0 | 80.0 | 79.8 | 76.5 | 84.7 | 78.5 | 76.4 |
| UEO | 92.3 | 75.5 | 90.4 | 82.9 | 72.7 | 73.5 | 59.3 | 64.7 | 86.9 | 78.3 | 85.1 | 82.3 | 80.3 | 83.7 | 76.0 | 85.7 | 78.1 | 78.6 |
| UEO(O) | 95.9 | 83.2 | 89.9 | 83.4 | 71.9 | 77.5 | 50.7 | 72.3 | 83.3 | 89.4 | 82.5 | 88.1 | 77.8 | 86.9 | 74.1 | 91.3 | 74.0 | 88.0 |
| *Prompt Only ⇈, Prompt + Normalization Layers⇊* | | | | | | | | | | | | | | | | | | |
| CLIP (Radford et al., 2021) | 88.2 | 81.2 | 89.5 | 82.1 | 70.1 | 74.9 | 55.3 | 64.7 | 84.2 | 78.0 | 83.1 | 82.5 | 76.6 | 84.8 | 73.6 | 85.6 | 69.5 | 77.9 |
| UPL (Huang et al., 2022) | 88.3 | 81.3 | 89.8 | 82.4 | 71.7 | 75.7 | 60.2 | 66.5 | 86.5 | 80.0 | 85.7 | 84.5 | 80.7 | 79.5 | 75.8 | 87.9 | 73.7 | 79.5 |
| POUF (Tanwisuth et al., 2023) | 93.7 | 74.7 | 89.9 | 82.7 | 71.7 | 75.7 | 61.1 | 65.8 | 85.2 | 78.0 | 85.4 | 82.2 | 80.2 | 81.7 | 75.7 | 86.9 | 75.1 | 79.9 |
| DANCE (Saito et al., 2020) | 89.5 | 73.6 | 33.3 | 36.7 | 70.4 | 75.1 | 58.2 | 64.3 | 84.8 | 79.8 | 84.6 | 83.2 | 79.9 | 84.7 | 73.6 | 85.6 | 70.1 | 77.8 |
| EntMin (Wang et al., 2021) | 89.4 | 80.2 | 90.3 | 83.0 | 70.7 | 75.0 | 56.2 | 65.0 | 85.1 | 78.5 | 83.4 | 82.6 | 77.6 | 84.8 | 73.8 | 85.7 | 69.6 | 77.9 |
| InfoMax (Liang et al., 2020) | 96.0 | 76.8 | 89.0 | 78.1 | 72.9 | 72.2 | 61.1 | 64.5 | 88.0 | 79.3 | 85.9 | 81.7 | 80.3 | 81.6 | 76.1 | 83.9 | 80.0 | 77.3 |
| UEO | 94.4 | 79.9 | 90.7 | 82.6 | 72.9 | 74.2 | 62.1 | 64.9 | 87.0 | 78.5 | 85.4 | 82.6 | 80.1 | 84.1 | 75.2 | 85.4 | 78.6 | 79.1 |
| UEO(O) | 96.6 | 85.5 | 90.2 | 85.2 | 71.8 | 78.9 | 54.2 | 74.8 | 86.0 | 88.4 | 84.7 | 88.1 | 77.3 | 87.6 | 73.2 | 90.8 | 74.3 | 86.5 |

Table 17: Results of different methods under the open-partial-set category shift (ResNet-50).

| Methods / Metrics (%) | VD (T) ACC | AUC | VD (V) ACC | AUC | OH (Ar) ACC | AUC | OH (Cl) ACC | AUC | OH (Pr) ACC | AUC | OH (Re) ACC | AUC | OF (A) ACC | AUC | OF (D) ACC | AUC | OF (W) ACC | AUC |
|---|---|---|---|---|---|---|---|---|---|---|---|---|---|---|---|---|---|---|
| CLIP (Radford et al., 2021) | 88.2 | 81.2 | 89.5 | 82.1 | 70.1 | 74.9 | 55.3 | 64.7 | 84.2 | 78.0 | 83.1 | 82.5 | 76.6 | 84.8 | 73.6 | 85.6 | 69.5 | 77.9 |
| UPL (Huang et al., 2022) | 88.7 | 81.0 | 90.0 | 82.9 | 71.4 | 74.9 | 57.2 | 63.0 | 85.5 | 79.4 | 84.5 | 83.1 | 78.4 | 84.3 | 73.7 | 85.6 | 72.0 | 77.7 |
| POUF (Tanwisuth et al., 2023) | 91.3 | 63.8 | 89.9 | 80.6 | 71.1 | 75.2 | 58.0 | 64.4 | 85.3 | 80.4 | 85.2 | 83.9 | 79.1 | 82.5 | 74.3 | 86.1 | 72.5 | 78.9 |
| DANCE (Saito et al., 2020) | 85.4 | 69.3 | 86.1 | 78.8 | 70.7 | 74.8 | 57.4 | 62.7 | 85.6 | 80.1 | 84.3 | 83.4 | 78.7 | 85.1 | 73.8 | 85.6 | 70.1 | 78.0 |
| EntMin (Wang et al., 2021) | 87.9 | 80.4 | 89.9 | 82.4 | 70.3 | 74.8 | 54.5 | 65.1 | 84.3 | 77.9 | 83.3 | 82.5 | 76.6 | 84.8 | 73.6 | 85.6 | 69.5 | 77.9 |
| InfoMax (Liang et al., 2020) | 88.3 | 64.1 | 90.6 | 71.2 | 71.7 | 71.9 | 57.8 | 65.3 | 87.1 | 78.9 | 85.4 | 82.5 | 80.1 | 80.3 | 74.2 | 85.5 | 74.9 | 78.9 |
| UEO | 85.4 | 68.7 | 89.6 | 80.8 | 71.8 | 74.3 | 59.0 | 64.4 | 86.1 | 79.9 | 85.6 | 83.2 | 79.9 | 84.3 | 74.6 | 86.0 | 74.7 | 79.3 |
| UEO(O) | 86.5 | 73.4 | 83.1 | 80.7 | 72.1 | 77.2 | 52.0 | 72.0 | 84.9 | 86.3 | 82.8 | 86.1 | 75.3 | 86.7 | 71.6 | 90.6 | 70.9 | 85.7 |
| *Prompt Only ⇈, Prompt + Normalization Layers⇊* | | | | | | | | | | | | | | | | | | |
| CLIP (Radford et al., 2021) | 88.2 | 81.2 | 89.5 | 82.1 | 70.2 | 74.9 | 55.3 | 64.7 | 84.2 | 78.0 | 83.1 | 82.5 | 76.6 | 84.8 | 73.6 | 85.6 | 69.5 | 77.9 |
| UPL (Huang et al., 2022) | 89.6 | 82.1 | 90.0 | 82.8 | 71.2 | 75.6 | 58.4 | 64.6 | 85.6 | 79.8 | 84.3 | 83.3 | 79.2 | 84.0 | 74.6 | 85.7 | 72.1 | 77.8 |
| POUF (Tanwisuth et al., 2023) | 94.1 | 67.7 | 89.1 | 81.4 | 71.7 | 75.5 | 60.3 | 65.7 | 85.4 | 78.9 | 85.1 | 83.3 | 78.7 | 82.3 | 75.1 | 85.9 | 72.5 | 79.0 |
| DANCE (Saito et al., 2020) | 86.9 | 65.7 | 90.0 | 83.9 | 70.3 | 75.0 | 59.3 | 64.7 | 84.6 | 78.4 | 84.3 | 82.9 | 77.8 | 84.4 | 73.5 | 85.6 | 69.4 | 77.6 |
| EntMin (Wang et al., 2021) | 88.2 | 78.5 | 90.1 | 82.5 | 70.4 | 74.9 | 56.1 | 65.4 | 85.1 | 78.2 | 83.3 | 82.5 | 76.9 | 84.8 | 73.8 | 85.6 | 69.5 | 77.9 |
| InfoMax (Liang et al., 2020) | 94.0 | 71.1 | 87.2 | 78.1 | 72.5 | 74.9 | 60.7 | 63.0 | 87.6 | 78.9 | 85.4 | 81.7 | 80.1 | 80.3 | 75.0 | 85.2 | 76.4 | 78.9 |
| UEO | 94.0 | 79.8 | 89.9 | 82.5 | 72.3 | 75.0 | 61.2 | 64.9 | 86.8 | 79.1 | 85.6 | 83.4 | 80.2 | 84.5 | 74.9 | 85.8 | 76.5 | 79.4 |
| UEO(O) | 93.9 | 75.6 | 85.4 | 81.4 | 71.5 | 78.4 | 54.4 | 74.2 | 86.0 | 85.9 | 84.7 | 87.1 | 75.6 | 87.0 | 72.1 | 90.4 | 72.3 | 86.6 |

