# OpenReview forum: "Towards Realistic Unsupervised Fine-tuning with Vision-Language Models"
_ICLR.cc/2024/Conference — Submitted to ICLR 2024_

### Official Review · Reviewer_spW5 · 2023-10-20

**Soundness:** 3 good
**Presentation:** 3 good
**Contribution:** 3 good
**Rating:** 6
**Confidence:** 4

**Summary:**

This paper introduces Universal Entropy Optimization (UEO), a method for unsupervised universal fine-tuning of vision-language models (VLMs) like CLIP. UEO aims to enhance the model's performance in two key aspects: accurate classification of samples from known classes and effective identification of samples from classes not present in the predefined classes. It does this by leveraging sample-level confidence and entropy optimization to handle out-of-distribution (OOD) samples. The paper presents results from experiments conducted across 15 domains, demonstrating that UEO outperforms baseline methods in terms of both generalization and OOD detection.

**Strengths:**

**Originality**

The paper proposes a simple yet efficient solution to address a unique and realistic setting of unsupervised universal fine-tuning. According to the authors, this is the first paper to tackle this practical setting. While the key principle of the method is similar to DANCE, unlike DANCE, the proposed approach does not require hyper-parameter selections, which can be challenging in the unsupervised setting.

**Quality**

This paper exhibits notable strengths in its hyper-parameter-free approach, Universal Entropy Optimization (UEO), which addresses the challenging task of unsupervised fine-tuning of vision-language models (VLMs) under real-world conditions, including potential out-of-distribution (OOD) samples in unlabeled data. Through comprehensive experiments conducted across diverse domains and the introduction of novel evaluation metrics like the AUC score, the paper showcases the effectiveness of UEO in both in-distribution classification and OOD detection. UEO's parameter-efficient methodology and emphasis on real-world scenarios make it a valuable contribution, marking its quality in the field of VLMs and unsupervised fine-tuning.


**Clarity**

The paper is well-written and easy to follow. The motivation of the paper is clear. The authors build upon previous works and cite them appropriately.

**Significance**

The proposed setting is practical. The paper tackles the problem where unknown classes can be present in the unlabeled data, replicating real-world scenarios. The analysis is thorough and the experiments across multiple settings show consistent improvements.

**Weaknesses:**

One potential weakness of the paper is that it relies on sample-level confidence weights to approximate entropy minimization and maximization. While this approach is innovative, it may be sensitive to the distribution of confidences within the unlabeled data. If the confidences are not well-calibrated or vary significantly across samples, it could affect the effectiveness of UEO. The performance of UEO might be influenced by the quality and reliability of the confidence estimates, and if the confidence estimates are noisy or inaccurate, it could lead to suboptimal results.

One experiment that is missing would involve applying the method to VLMs that are not well-calibrated and observing its impact on the performance of the fine-tuned model.

Another potential drawback is that, when compared to InfoMax, there does not appear to be a significant improvement in accuracy on certain datasets. For instance, on the DomainNet dataset, there is no noticeable difference in performance in accuracy.

**Questions:**

1. Can the universal entropy optimization loss be potentially incorporated during the contrastive pre-training phase as well?

2. Apart from being hyper-parameter free, could the author elaborate more on the benefit of using the proposed universal entropy optimization instead of the entropy separation loss in DANCE?

3. Since the method relies on the confidence of the model, could the authors discuss the effect of calibration of predicted probabilities on the performance of the method?

I would be willing to raise my score if the authors could address my concerns and answer the questions above.

---

> ### Author Response · Authors · 2023-11-16
>
> We thank the reviewer for providing valuable comments and address your concerns in the following responses.
>
> > **[Q1]**. While this approach is innovative, it may be sensitive to the distribution of confidences within the unlabeled data. If the confidences are not well-calibrated or vary significantly across samples, it could affect the effectiveness of UEO.
>
> **[A1]**. We agree on the significance of the instance weight $w(x)$. However, UEO has been empirically demonstrated to be not sensitive to confidences. To assess the calibration of instance weights, we utilize the AUC score as an indicator. For instance, when comparing the results on the DN (Avg.) dataset under open-partial shift with two different backbones (ResNet-50 in Table 4 and ViT-B/16 in Table 5), UEO consistently outperforms the CLIP baseline in terms of the accuracy (ACC) score. It is worth noting that improved calibration may not necessarily result in larger gains. Similar observations can be made when comparing different datasets under the same backbone.
>
> |  DN (Avg.) [open-partial]    | ACC  | AUC  |
> |----------------|------|------|
> | CLIP (ResNet-50) | 47.3 | 66.5 |
> | UEO (ResNet-50)  | 51.9 | 67.4 |
> | CLIP (ViT-B/16) | 58.2 | 72.6 |
> | UEO (ViT-B/16)  | 62.0 | 72.9 |
>
> > **[Q2]**. One experiment that is missing would involve applying the method to VLMs that are not well-calibrated and observing its impact on the performance of the fine-tuned model.
>
> **[A2]**. As obtaining a poorly calibrated VLM is challenging, we kindly refer the reviewer to **[A5] @ Reviewer eDVD** for a visual pre-trained source model (VM). Carefully comparing the gains of UEO over the initized model in two following tables, we find that the AUC scores vary a lot across different models, and UEO always increases in terms of the ACC score.
>
> |  OF(A) [open-partial]  | ACC  | AUC  |
> |----------------|------|------|
> | VM | 65.2 | 71.8 |
> | UEO(VM)| 70.2 |	74.1 |
> | CLIP| 76.6 | 84.8 |
> | UEO(CLIP)  | 80.2 | 84.5 |
>
> |  OH(A) [open-partial]  | ACC  | AUC  |
> |----------------|------|------|
> | VM | 54.5	|64.5 |
> | UEO(VM)| 58.2 |	64.7 |
> | CLIP| 70.2 |74.9 |
> | UEO(CLIP)  | 72.3 | 75.0 |
>
> > **[Q3]**. When compared to InfoMax, there does not appear to be a significant improvement in accuracy on certain datasets. For instance, on the DomainNet dataset, there is no noticeable difference in performance in accuracy.
>
> **[A3]**. We respectfully disagree. As stated in **[A2] @ Reviewer pTa8**, UEO consistently outperforms InfoMax when considering both accuracy (ACC) and area under the curve (AUC) together. While the difference between UEO and InfoMax on the DomainNet dataset may appear small, UEO consistently exhibits superior AUC scores under different category shifts and backbones.
>
> > **[Q4]**. Can the universal entropy optimization loss be potentially incorporated during the contrastive pre-training phase as well?
>
> **[A4]**. That sounds like a reasonable and interesting direction for future work. When we have paired text-image pairs along with some unpaired images during the pre-training phase, we can take the texts from the pairs as pre-defined class names and incorporate a simialr loss on these unpaired images with the contrastive pre-training loss over image-text pairs.
>
> > **[Q5]**. Apart from being hyper-parameter free, could the author elaborate more on the benefit of using the proposed universal entropy optimization instead of the entropy separation loss in DANCE?
>
> **[A5]**. In **[A4] @ Reviewer eDVD**, we show the disadvatnages of thresholding-based methods. Thresholding-based methods, including the entropy separation loss in DANCE, indeed rely heavily on the chosen threshold, and their performance can be sensitive to the absence of OOD samples, especially in scenarios like closed-set and partial-set category shifts where OOD samples are not present. Mistakenly classifying hard ID samples as OOD samples can introduce risks and impact the overall effectiveness of these methods.

---

> > ### Comment · Reviewer_spW5 · 2023-11-19
> > **Rebuttal response**
> >
> > Based on the author's response and other reviewers' comments, I decided to increase my score.

---

> > > ### Author Response · Authors · 2023-11-19
> > >
> > > Thanks for your responses and we appreciate your endorsement.

---

### Official Review · Reviewer_pTa8 · 2023-10-27

**Soundness:** 1 poor
**Presentation:** 3 good
**Contribution:** 3 good
**Rating:** 5
**Confidence:** 4

**Summary:**

This paper introduces a novel task setting for unsupervised CLIP fine-tuning, where the label spaces of unlabeled data and predefined text classes are partially overlapped. As a result, the trained model is required to concurrently detect out-of-distribution categories while recognizing samples within the predefined classes. To address this challenge, the paper proposes a straightforward approach that aims to minimize the conditional entropy of confident samples and maximize the marginal entropy of less confident ones. Experiments are performed on benchmark datasets.

**Strengths:**

1. To the best of my knowledge, the proposed task setting is novel. I believe it offers a valuable direction for unsupervised CLIP adaptation.
2. The proposed approach is straightforward and results in a general improvement.
3. Experiments are carried out on widely accepted DA benchmarks.

**Weaknesses:**

1. As highlighted in the introduction and method sections, the paper's primary focus is the class discrepancy between unlabeled data and the predefined label space. However, the principal experiments are based on domain adaptation datasets, characterized predominantly by distributional differences between domains. I believe more general classification datasets (e.g., ImageNet and SUN397 as in CoOp) should be employed to define the task setting and verify the efficacy of the proposed method.
2. The introduced method bears a significant resemblance to existing mutual information maximization losses. The sole distinction appears to be the instance weight, which is based on maximum prediction probability. Additionally, the performance gains seem rather marginal when compared with its peers.
3. Missing ablation studies. The methodology encompasses two critical components: firstly, the entropy-related training objective, and secondly, the parameter-efficient tuning. I'm uncertain if the competing methods in Tab. 1 also implement the same parameter-efficient tuning. Furthermore, the prompt and affine parameters of the parameter-efficient tuning should be dissected and examined individually.
4. Missing references. Some related studies delve into various task settings, such as black-box DA in [1], partial DA in [2], and universal DA in [3].

[1] Unsupervised Domain Adaptation of Black-Box Source Models, BMVC21
[2] Universal domain adaptation, CVPR19
[3] Partial adversarial domain adaptation, ECCV18

**Questions:**

See weaknesses.

---

> ### Author Response · Authors · 2023-11-16
>
> We thank the reviewer for providing valuable comments and address your concerns in the following responses.
>
> > **[Q1]**. I believe more general classification datasets (e.g., ImageNet and SUN397 as in CoOp) should be employed to define the task setting and verify the efficacy of the proposed method.
>
> **[A1]**. A great advice. To validate the effectiveness of the proposed method (UEO), we additionally utilize three widely recognized classification datasets (i.e., ImageNet, SUN397, and Food101) and present the results under four category shifts in the tables below.
>
> | ImageNet | ACC (C) | AUC (C) | ACC (P) | AUC (P) | ACC (O) | AUC (O) | ACC (OP) | AUC (OP) | ACC (Avg.) | AUC (Avg.) |
> |---------|---------|---------|---------|---------|---------|---------|---------|---------|---------|---------|
> | CLIP    | 63.8 | 73.8 | 63.8 | 73.8 | 63.8 | 73.8 | 63.8 | 73.8    | 63.8 | 73.8 |
> | UPL     | 65.6 | 73.6 | 65.2 | 73.9 | 65.8 | 73.5 | 65.6 | 72.9    | 65.5 | 73.5 |
> | POUF    | 65.7 | 72.0 | 65.2 | 72.8 | 65.7 | 71.0 | 65.8 | 71.7    | 65.6 | 71.9 |
> | DANCE   | 64.7 | 72.8 | 64.3 | 73.3 | 64.9 | 72.1 | 64.4 | 72.6    | 64.6 | 72.7 |
> | EntMin  | 64.0 | 73.5 | 64.1 | 73.8 | 63.9 | 73.1 | 63.8 | 73.3    | 63.9 | 73.4 |
> | InfoMax | 66.3 | 71.8 | 65.4 | 73.4 | 65.1 | 64.5 | 65.3 | 68.3    | 65.5 | 69.5 |
> | UEO     | 65.9 | 73.0 | 65.3 | 74.2 | 65.9 | 72.4 | 65.9 | 73.2    | 65.7 | 73.2 |
>
> | SUN397 | ACC (C) | AUC (C) | ACC (P) | AUC (P) | ACC (O) | AUC (O) | ACC (OP) | AUC (OP) | ACC (Avg.) | AUC (Avg.) |
> |---------|---------|---------|---------|---------|---------|---------|---------|---------|---------|---------|
> | CLIP    | 63.7 | 68.6 | 63.7 | 68.6 | 63.7 | 68.6 | 63.7 | 68.6    | 63.7 | 68.6 |
> | UPL     | 66.0 | 69.0 | 65.6 | 69.0 | 66.2 | 69.1 | 66.0 | 69.0    | 65.9 | 69.0 |
> | POUF    | 66.2 | 69.0 | 65.9 | 69.0 | 66.2 | 68.9 | 66.1 | 69.0    | 66.1 | 69.0 |
> | DANCE   | 64.8 | 68.9 | 64.6 | 68.8 | 64.9 | 68.9 | 64.8 | 68.8    | 64.8 | 68.8 |
> | EntMin  | 64.1 | 68.7 | 64.0 | 68.6 | 64.1 | 68.7 | 64.1 | 68.6    | 64.1 | 68.6 |
> | InfoMax | 67.0 | 68.7 | 66.8 | 68.8 | 67.0 | 68.5 | 67.0 | 68.6    | 67.0 | 68.7 |
> | UEO     | 66.5 | 68.9 | 66.4 | 69.0 | 66.7 | 68.8 | 66.6 | 68.8    | 66.5 | 68.9 |
>
> | Food101 | ACC (C) | AUC (C) | ACC (P) | AUC (P) | ACC (O) | AUC (O) | ACC (OP) | AUC (OP) | ACC (Avg.) | AUC (Avg.) |
> |---------|---------|---------|---------|---------|---------|---------|---------|---------|---------|---------|
> | CLIP    | 79.7 | 77.6 | 79.7 | 77.6 | 79.7 | 77.6 | 79.7 | 77.6    | 79.7 | 77.6 |
> | UPL     | 80.2 | 77.7 | 80.6 | 77.3 | 80.3 | 78.0 | 81.2 | 78.4    | 80.6 | 77.8 |
> | POUF    | 81.5 | 77.6 | 81.1 | 77.6 | 81.5 | 77.2 | 81.3 | 77.2    | 81.3 | 77.4 |
> | DANCE   | 81.0 | 77.4 | 80.4 | 77.4 | 80.4 | 76.3 | 80.0 | 76.2    | 80.4 | 76.8 |
> | EntMin  | 80.1 | 77.6 | 79.9 | 77.6 | 80.0 | 77.2 | 79.9 | 77.1    | 80.0 | 77.4 |
> | InfoMax | 82.5 | 77.1 | 82.1 | 77.7 | 82.6 | 76.5 | 82.3 | 76.0    | 82.4 | 76.8 |
> | UEO     | 82.4 | 77.9 | 81.9 | 77.9 | 82.3 | 77.4 | 82.1 | 77.7    | 82.2 | 77.7 |
>
> [(C): closed-set, (P): partial-set, (O): open-set, (OP): open-partial]
>
> On the ImageNet dataset, UEO and UPL are top two methods when considering both ACC and AUC simultaneously. Similarly, UEO and InfoMax are top two methods on the SUN397 dataset. Notably, UEO achieves the best performance on the FOOD101 dataset. In summary, UEO consistently attains the best or competitive performance across different datasets, while other methods exhibit more variability across the datasets.

---

> ### Author Response · Authors · 2023-11-16
>
> > **[Q2]**. The introduced method bears a significant resemblance to existing mutual information maximization losses. Additionally, the performance gains seem rather marginal when compared with its peers.
>
> **[A2]**. We acknowledge that a direct comparison of the objective in Equation (4) with InfoMax [Liang et al., ICML 2020] reveals the sole difference lies in the instance weight $w(x)$. However, it is crucial to note that the motivation behind this choice is entirely different. We explain the relaxation from Eq. (3) to Eq. (4) in **[A3] @ Reviewer eDVD**. Concerning the performance gains, we collect the average results of InfoMax and UEO from Tables 1-4 in the table below. It appears that UEO outperforms InfoMax in a majority of cases, with only a few cases where it is inferior to InfoMax.
>
> |         | DN   | DN   | VD   | VD   | OH   | OH   | OF   | OF   |
> |---------|------|------|------|------|------|------|------|------|
> | (C)     | ACC  | AUC  | ACC  | AUC  | ACC  | AUC  | ACC  | AUC  |
> | InfoMax | 52.0 | 66.3 | 92.6 | 81.4 | 77.0 | 75.8 | **79.2** | 82.5 |
> | UEO     | 51.9 | **67.2** | 92.2 | **84.6** | 76.8 | 75.6 | 78.1 | **84.1** |
> | (P)     | ACC  | AUC  | ACC  | AUC  | ACC  | AUC  | ACC  | AUC  |
> | InfoMax | 51.7 | 66.4 | 88.7 | 79.4 | 76.3 | 75.4 | 76.7 | 83.6 |
> | UEO     | 51.8 | **67.2** | **90.0** | **82.9** | **76.6** | 76.3 | 76.4 | 83.4 |
> | (O)     | ACC  | AUC  | ACC  | AUC  | ACC  | AUC  | ACC  | AUC  |
> | InfoMax | 51.9 | 65.8 | 92.5 | 77.5 | 77.0 | 74.4 | **78.8** | 80.9 |
> | UEO     | 51.9 | **67.4** | 92.6 | **81.2** | 76.8 | 75.0 | 77.9 | **82.9** |
> | (OP)    | ACC  | AUC  | ACC  | AUC  | ACC  | AUC  | ACC  | AUC  |
> | InfoMax | 51.9 | 66.0 | 90.6 | 74.6 | 76.5 | 73.9 | 77.2 | 81.5 |
> | UEO     | 51.9 | **67.4** | **92.0** | **81.2** | 76.5 | **75.6** | 77.2 | **83.2** |
>
> [(C): closed-set, (P): partial-set, (O): open-set, (OP): open-partial]
>
> > **[Q3]**. I'm uncertain if the competing methods in Tab. 1 also implement the same parameter-efficient tuning strategy. Furthermore, the prompt and affine parameters of the parameter-efficient tuning should be dissected and examined individually.
>
> **[A3]**. We feel sorry for any confusion. As written in the last paragrpah in Sec 4.1, all the methods are implmented with the same parameter-efficient tuning techique. In particular, we reduce the learning rate for EntMin due to its rapidly decreasing results.
>
> To verify the effectiveness of the proposed parameter-efficient tuning strategy, we present results with only text prompt tuning in the Appendix (Tables 14-17). Additionally, we investigate the influence of **different combinations of parameters during the fine-tuning process in Fig. 3**. In these figures, the first two columns denote optimizing the text prompt and visual affine parameters, respectively.
>
> > **[Q4]**. Missing references [1,2,3] about domain adaptation?
>
> **[A4]**. Thanks for your kind reminder. [r1] explores the model adaptation problem in a black-box manner, [r2] considers domain adaptation under open-partial category shift, and [r3] addresses domain adaptation under partial-set category shift. These papers represent various domain adaptation problem setups, and we will include them in the related works section. Given that these methods are tailored to specific category shifts, which significantly differ from the problem setup presented here, direct comparisons with UEO in the experiments prove challenging.
>
> - [r1] Zhang, Haojian, et al. "Unsupervised Domain Adaptation of Black-Box Source Models." In Proc. BMVC 2021.
> - [r2] You, Kaichao, et al. "Universal domain adaptation." Proc. CVPR. 2019.
> - [r3] Cao, Zhangjie, et al. "Partial adversarial domain adaptation." Proc. ECCV. 2018.

---

> > ### Comment · Reviewer_pTa8 · 2023-11-20
> > **Rebuttal response**
> >
> > Thank you for your response. Your additional experiments on general datasets have been well received.  I have decided to increase my score slightly. While I appreciate the new task setting, my concerns regarding your proposed method have not been addressed. Firstly, as you acknowledged, the only difference between this method and InfoMax is the instance weight, which limits its technical contributions. Secondly, the improvement of this method compared to the baseline (e.g., CLIP, UPL) is relatively marginal, especially when evaluated on widely recognized large-scale datasets such as ImageNet.

---

> > > ### Author Response · Authors · 2023-11-20
> > > **Thank you for your endorsement and the increased score.**
> > >
> > > Thank you for increasing the score, and we are pleased to address the remaining concerns.
> > >
> > > 1. It is indeed acknowledged that UEO and InfoMax exhibit slight differences in instance weight, but rooted in distinct motivations. InfoMax is typically applied in closed-set category shifts, aiming to minimize the entropy of each instance while expecting a uniform class distribution across all classes to prevent class collapse in unsupervised learning. On the other hand, UEO is designed to minimize the entropy of potentially ID samples and maximize the entropy of potentially OOD samples, making it more suitable for unsupervised learning in diverse category shifts.
> > >
> > > 2. The comprehensive results presented in Tables 1-4, showcasing UEO's outperformance against CLIP and UPL across various category shifts, reinforce its efficacy. Concerning the results in additional large-scale general datasets, despite a minor decrease in AUC score on ImageNet, UEO consistently maintains AUC scores and improves ACC scores across all three large datasets (ImageNet, SUN397, Food101) compared to CLIP. Similar favorable outcomes are observed in the comparison with UPL.
> > >
> > > In summary, we believe that this proposed simple method, UEO, has the potential to establish a stronger baseline for the new task setting. Your support and feedback are greatly valued.

---

> > > ### Author Response · Authors · 2023-11-23
> > > **End of discussion approaching**
> > >
> > > Dear Reviewer,
> > >
> > > Since the discussion deadline is approaching in less than 12 hours, we kindly request your feedback on whether the follow-up response adequately addresses your concerns. If you have any more questions, we would be happy to provide further clarification.
> > >
> > > Your timely response is greatly appreciated.
> > >
> > > Thanks.

---

> > > ### Author Response · Authors · 2023-11-23
> > > **Your Feedback is Appreciated in the Last Minute.**
> > >
> > > Dear reviewer,
> > >
> > > Thanks again for your endorsement and the increased score.
> > >
> > > Regarding the two remaining concerns highlighted in your recent response, we seek confirmation on whether our clarifications have satisfactorily addressed them. We would greatly appreciate your feedback before the end of the rebuttal phase.
> > >
> > > The authors

---

### Official Review · Reviewer_tymK · 2023-10-30

**Soundness:** 3 good
**Presentation:** 2 fair
**Contribution:** 3 good
**Rating:** 8
**Confidence:** 3

**Summary:**

This paper proposes a novel setting called "unsupervised universal fine-tuning," which involves both in-distribution prediction and out-of-distribution detection. To tackle this problem, the authors presented an approach called "universal entropy optimization." It utilizes the confidence of each sample to minimize the entropy of confident samples but maximize the entropy of confident samples. These combined lead to improvement for both generalization and out-of-distribution detection on benchmarks like DomainNet, VISDA-V, Office-OF, etc.

**Strengths:**

* The problem setup of "unsupervised universal finetuning" seems reasonable and is grounded in the disadvantages of previous settings.
* I think the approach of "universal entropy optimization" (UEO), especially Eqn. 3, is interesting in achieving maximization and minimization at the same time. I don't directly work in this field and am not sure whether Eqn. 3 has been used by other people. Nonetheless, I think the UEO approach in the paper is intriguing.
* The performance demonstrated in the experiment section supports the effectiveness of the approach.

**Weaknesses:**

(Details in the questions section) I think the authors might need to clarify several questions to fully illustrate their novel explorations, including the role of vision-language models and the significance of the new setting. Additionally, the numbers in Table 1-4 are quite close in some datasets (though I admire and appreciate the exhaustive evaluation from the authors), so it is also better to make more clarifications.

**Questions:**

1. What is the special role of the "vision-language model" in the paper or the investigated problem? It seems to me the approach and problem-setting are applicable to models beyond CLIP?

2. Following the above question, I think the authors need to better clarify how their experiment setting differs from previous works. Specifically, the authors mentioned "unsupervised universal fine-tuning" as a novel fine-tuning setup, but it seems the evaluation directly adopted the previous datasets without special curation. Therefore, I am wondering if this is a new setting, or some previous setting adapted to CLIP, or some other cases?

3. The numbers in the tables are quite close for some datasets, and the performance for UEO is not the best on some datasets, such as  the avg numbers. Therefore, I think clarifications on the following questions would be helpful:
* Is there a clear baseline of the UEO approach, e.g., some simple modification or fine-tuning strategy to CLIP for this setting?
* What is the variance of these numbers?
* Which is the largest and hardest dataset?
* State-of-the-art is not necessary for me, but the authors might need to investigate more into the difference in performance and offer some insights. Let's take Table 1 for example, the gap between UEO and UPL on OH (Avg.) in quite significant, what might be the cause?

---

> ### Author Response · Authors · 2023-11-16
>
> We appreciate the reviewer for providing valuable comments, and we address your concerns in the following responses.
>
> > **[Q1]**. What is the special role of the "vision-language model" in the paper or the investigated problem? It seems to me the approach and problem-setting are applicable to models beyond CLIP?
>
> **[A1]**. Firstly, our approach aligns with established practices in prior studies, including references [r1, r2], where we explore the application of vision-language models with a specific focus on the CLIP model. Secondly, it is true that our method and the associated problem setup remain reasonable even when utilizing other pre-trained networks, such as traditional model adaptation. We present the corresponding results in **[A5] @ Reviewer eDVD**. However, the utilization of vision-language models like CLIP proves more apt for the proposed Unsupervised Universal Fine-Tuning problem. This is attributed to the advantage of not requiring labeled data collection efforts for training, thanks to the pre-defined list of class names.
>
> - [r1]. Zhou, Kaiyang, et al. "Learning to prompt for vision-language models." International Journal of Computer Vision 130.9 (2022): 2337-2348.
> - [r2]. Du, Yu, et al. "Learning to prompt for open-vocabulary object detection with vision-language model." Proceedings of the IEEE/CVF Conference on Computer Vision and Pattern Recognition. 2022.
>
> > **[Q2]**. I think the authors need to better clarify how their experiment setting differs from previous works. I am wondering if this is a new setting, or some previous setting adapted to CLIP, or some other cases?
>
> **[A2]**. We feel sorry for any confusion. We would like to clarify that the proposed unsupervised universal fine-tuning problem **is not derived from other related topics but constitutes a new and intriguing task**. Since the proposed task is closely related to several tasks in the domain adaptation area (i.e., model adaptation and universal domain adaptation), we employ several popular benchmarks (e.g., Office, OfficeHome, and DomainNet) in the domain adaptation area. We provide additional results on more general benchmarks in **[A1] @ Reviewer pTa8**.
>
> To the best of our knowledge, our task represents the first exploration into both ID generalization and OOD detection for unsupervised fine-tuning under unknown category shifts. Unlike most prior domain adaptation methods that assume knowledge of the category shift in advance and predominantly emphasize the ability of ID generalization, our task is more realistic. In the context of universal domain adaptation tasks, where the target data for adaptation and the test data share the same label space, evaluating the ability of OOD detection becomes challenging (as depcited in Evaluation of Sec. 3.2). Instead, our task introduces a distinct fine-tuning and evaluation protocol, along with different evaluation metrics.
>
> > **[Q3]**. Additionally, the numbers in Table 1-4 are quite close in some datasets (though I admire and appreciate the exhaustive evaluation from the authors), so it is also better to make more clarifications. (four questions ...)
>
> **[A3]**. For each question, we offer a point-to-point response below.
>
> - For the proposed parameter-tuning stragey, we show the results of only prompt tuning in Fig. 3 and Tables 14 - 17 in the Appendix. Regarding the objectives, EntMin and InfoMax could be considered as two baselines for UEO. Further comparisons with two-step thresholding variants are elaborated in **[A4] @ Reviewer eDVD**.
> - Regarding the variance of the numbers, we conducted all the experiments using different seeds and observed a relatively small variance, approximately 0.1 for both ACC and AUC. Therefore, we omitted the variance in these tables
> - Generally, DomainNet could be considered as the largest and hardest dataset in this paper. For instance, the Real domain in the DomainNet dataset comprises approximately 120k images across 345 classes. Notably, the ACC and AUC scores of the original CLIP model are consistently the lowest among all the datasets.
> - In some datasets, UEO might not exhibit a clear superiority over the baselines, primarily for two reasons. Firstly, unsupervised fine-tuning is inherently challenging, and the majority of fine-tuning methods do not significantly outperform the original CLIP model. Secondly, enhancing both ACC and AUC scores concurrently is a difficult task. In most situations, a trade-off exists between these two scores. Notably, UEO consistently improves the ACC score without compromising the AUC score, outperforming other methods across the majority of datasets and category shifts.

---

> ### Author Response · Authors · 2023-11-21
> **Invitation to further discussion**
>
> Dear reviewer,
>
> We genuinely appreciate the time and effort you've invested in reviewing our paper. We have carefully provided relevant responses and results to your concerns. We are eager to further discuss with you and gain your insights **before the end of the Author/Reviewer phase**. Please let us know if any aspect of our work remains unclear or if you have additional feedback.
>
> Thank you.

---

> > ### Comment · Reviewer_tymK · 2023-11-21
> >
> > Thank you for your clarification and detailed answers! I think this paper has done a lot of work and would be meaningful to the researchers in the related fields. I suggest adding the clarified points above into the paper to better illustrate the main message. I don't have further follow-up questions for now.

---

> > > ### Author Response · Authors · 2023-11-21
> > > **Thanks for your endorsement.**
> > >
> > > We're grateful for your appreciation and endorsement. Your review holds significant value for us, and we'll ensure to incorporate the clarified points into the revision.

---

### Official Review · Reviewer_eDVD · 2023-10-31

**Soundness:** 2 fair
**Presentation:** 3 good
**Contribution:** 2 fair
**Rating:** 5
**Confidence:** 4

**Summary:**

This study tackles the problem of finetuning a vision-language model like CLIP on new unlabeled data with samples of unknown classes. To this end, a new approach called universal entropy optimization (UEO) is proposed. UEO utilizes the CLIP output score with known classes to determine whether a sample is an out-of-distribution (OOD) one. Then, the in-distribution samples are optimized following the standard entropy minimization strategy whereas the OOD samples are forced to maximize their prediction entropy. This finetuning process is parameter efficient as only the text prompts are involved during finetuning. Results using various methods across different open-set finetuning scenarios are evaluated, and the proposed strategy is validated to be effective.

**Strengths:**

- The attacked problem of the side effect caused by out-of-distribution samples during unsupervised finetuning is interesting, as it is encountered for many downstream applications of a large pretrained model.

- The effort of trying to adopt a unified adaptive loss function for both ID and OOD samples are appreciated, even though this goal is not quite accomplished in this study as would be later discussed.

- This method is validated on both ResNet and Vit-B backbones across various domain adaptation datasets, and comparisons with previous studies indicate a superior performance of the current method.

**Weaknesses:**

- The strategy of entropy minimization for ID samples and entropy maximization for OOD samples have been a popular method[1, 2, 3]. This study applies the principle to the field of vision-language models. Despite its effectiveness on different benchmarks, the core idea resembles traditional ones, which would compromise the novelty of this study.

- I understand that the authors contribute in a generalized form as in Eq. (3) & (4) for the loss function of both ID and OOD samples. However, a similar principle of maximizing Mutual Information ID samples and penalizing the mutual information of OOD samples has been also proposed in [4].

- The theoretical derivation of Eq. (3) & (4) could be more explicit and detailed. The current version appears to be intuitive and lack thorough theoretical analysis. Eq. (3) is proposed just to satisfy the rule that minimize the entropy ID instances and maximization the entropy of OOD samples”. However, no theoretical guarantee is provided so that Eq. (3) & (4) would always satisfy the above principle. The explanation is also missing of how Eq. (3) & (4) would be more suitable than a simple stepwise function, e.g. $L_{ID}=H(p(x)$ and $L_{OOD}=-H(p(x))$, and the determination of OOD samples follows the common practice as introduced in Sec. 3.1.

- As for the scope of application of the proposed method, it appears to be a general OOD method that can be also applied to traditional classification networks. I wonder why this method is applied to only CLIP method instead of extending it to other pretrained backbones.


References

1.Chan, R., Rottmann, M. & Gottschalk, H. Entropy Maximization and Meta Classification for Out-of-Distribution Detection in Semantic Segmentation. in *2021 IEEE/CVF International Conference on Computer Vision (ICCV)* 5108–5117 (IEEE, 2021).

2.Macêdo, D., Ren, T. I., Zanchettin, C., Oliveira, A. L. I. & Ludermir, T. Entropic Out-of-Distribution Detection. in *2021 International Joint Conference on Neural Networks (IJCNN)* 1–8 (2021).

3.Lee, K., Lee, H., Lee, K. & Shin, J. Training Confidence-calibrated Classifiers for Detecting Out-of-Distribution Samples. Preprint at [http://arxiv.org/abs/1711.09325](http://arxiv.org/abs/1711.09325) (2018).

4.Nimah, I., Fang, M., Menkovski, V. & Pechenizkiy, M. ProtoInfoMax: Prototypical Networks with Mutual Information Maximization for Out-of-Domain Detection. in *Findings of the Association for Computational Linguistics: EMNLP 2021* (eds. Moens, M.-F., Huang, X., Specia, L. & Yih, S. W.) 1606–1617 (Association for Computational Linguistics, 2021).

**Questions:**

From my point of view, Eq. (3) can also be viewed as an implicit threshold strategy to determine OOD samples. Specifically, assume of the max softmax probability $w$ follows a uniform assumption $w\sim \mathcal U(\frac{1}{C},1)$ , where $C$ denotes the total number of ID classes. The expectation $\mathbb E(w)=\frac{1}{2}(1-\frac{1}{C^2})$ and $\mathbb E(\frac 1 w)=log(C)$. Therefore, $\tilde w(x) - \tilde \Phi(w(x))\approx \frac{w}{\mathcal B_t \mathbb E(w)} - \frac{1/w}{\mathcal B_t \mathbb E(1/w)}$, and the thereshold for determining whether a sample is OOD now becomes $\lambda=\frac{1}{2} (1-\frac{1}{c^2})\log(C)$. In other words, Eq. (3) could be also one implicit form of thresholding strategy. I think the author should state the explicit benefit brought by the unified form as in Eq. (3) and (4) compared to a hard thresholding one. For example, we can observe in the form of $\lambda$ that $\lambda$ increases with the number of classes $C$ , yet I could not understand the rationale of this property.

---

> ### Author Response · Authors · 2023-11-16
>
> We thank the reviewer for providing valuable comments and address your concerns in the following responses.
>
> > **[Q1]**. The resemblance to previous methods [1,2,3] would compromise the novelty of this study.
>
> **[A1]**. After carefully reviewing these papers [1,2,3], we observe that each of them utilizes a similar strategy, aiming to minimize the entropy of ID samples and maximize the entropy of OOD samples to enhance OOD detection capability. While our method does share a similar high-level idea, it diverges from these approaches in two key aspects.
>
> **The first difference lies in the problem setup**, as our fine-tuning process is executed solely with unlabeled data, without explicit ID and OOD samples. In certain category shifts, such as closed-set and partial-set scenarios, OOD samples may not even exist in the unlabeled data, a condition that was not feasible for previous methods.
>
> **The second difference lies in the ultimate objective**. If we know $w(x)\to 1$ for ID samples and $w(x)\to 0$ for OOD samples for **unlabeled** data in advance, Eq. (3) exhibits similarities to prior OOD detection methods [1,2,3]. However, our proposed method optimizes the objective in Eq. (4), representing a more relaxed yet robust alternative to Eq. (3).
>
> > **[Q2]**. However, a similar principle of maximizing Mutual Information ID samples and penalizing the mutual information of OOD samples has been also proposed in [4].
>
> **[A2]**. Apologies for any confusion. Our method is different from [4]. Firstly, the method in [4] adopts the meta-learning framework that splits ID and OOD samples in the meta-training step, retaining the ground truth knowledge of ID and OOD during training. Secondly, the Mutual Information Maximization training objective in [4] is defined as the binary cross-entropy loss between ID and OOD prediction, which is distinct from our entropic loss. To be honest, minimizing the scores for ID samples and maximizing the scores for OOD samples a straightforward solution given the presence of ID and OOD samples in the training process. However, in the context of unsupervised fine-tuning, a key challenge arises in distinguishing ID from OOD samples. To circumvent the need for sensitive thresholding operations, we propose the relaxed optimization objective in Eq. (4).
>
> Beyond the unified objective in Eq. (4), our paper introduces **a novel unsupervised fine-tuning problem setup** that considers both ID generalization and OOD detection. This is especially crucial in handling various category shifts between the training data and the pre-defined label set, making our fine-tuning approach a notable contribution to the field.
> particularly in the face of various category shifts between the training data and the pre-defined label set. We consider such a fine-tuning problem setup to be a significant contribution to the field. Additionally, we provide **a new parameter-efficient fine-tuning strategy** for CLIP that optimizes not only the text prompts but also the normalization layers in the visual branch.
>
> > **[Q3]**. How do Eq. (3) and Eq. (4) theoretically satisfy the principle "minimize the entropy of ID instances and maximize the entropy of OOD samples"?
>
> **[A3]**. As explained above, when $w(x)\to 1$ for unlabeled ID samples and $w(x)\to 0$ for unlabeled OOD samples, Eq. (3) becomes the optimal objective that minimizes the entropy of ID instances and maximization the entropy of OOD samples. In the context of the unsupervised fine-tuning problem, where binary labels ("ID" or "OOD") are not available, we suggest employing the maximum softmax probability as an approximation for the ideal weight, as outlined in Eq. (3).
>
> As the estimated weight $w(x)$ may not always be accurate — for example, no OOD samples exist in the unlabeled data under the closed-set and partial-set category shifts - maximizing the entropy can be harmful. Intuitively, we could maximize the entropy of mean prediction over OOD samples instead of the average entropy of individual OOD samples, leveraging the inequality $H(\mathbb{E}(p(x))) \geq \mathbb{E}(H(p(x)))$. Honestly, optimizing Eq. (3) and Eq. (4) does not theoretically guarantee the principle in the challenging unsupervised fine-tuning problem, but we believe the proposed objective in Eq. (4) establishes a simple yet strong baseline, as evidenced in extensive experiments.

---

> ### Author Response · Authors · 2023-11-16
>
> > **[Q4]**. The explanation is also missing of how Eq. (3) & (4) would be more suitable than a simple stepwise function, e.g. $L_{ID}=H(p(x))$ and $L_{OOD}=-H(p(x))$, and the determination of OOD samples follows the common practice as introduced in Sec. 3.1.
>
> **[A4]**. Thanks for the advice. As written in the common practice in Sec. 3.1, $\lambda$ is chosen so that a high fraction of ID data (e.g., 90%) is above the threshold. Given the absence of explicit ID data in the unsupervised fine-tuning problem, we exclude it as a baseline method, denoted as **EO-ID**, in our experiments.
>
> Following the interesting view from the reviewer, we assume $w\sim \mathcal{U}(\frac{1}{C},1)$, then obtain $\mathbb{E}(w)=\frac{1-\frac{1}{C^2}}{2}\times\frac{1}{1-\frac{1}{C}}$ and $\mathbb{E}(\frac{1}{w})=log(C)\times\frac{1}{1-\frac{1}{C}}$. To discover the sample whose entropy is minimized in Eq. (3), we have $\hat{w}(x) - \hat{\Phi}(w(x))>0 \Rightarrow \frac{w}{\mathbb{E}(w)} > \frac{1/w}{\mathbb{E}(\frac{1}{w})} \Rightarrow w^2\times \mathbb{E}(\frac{1}{w})> \mathbb{E}(w) \Rightarrow w > \sqrt{\frac{1-\frac{1}{C^2}}{2log(C)}}$. By the way, the threshold $\lambda=\sqrt{\frac{1-\frac{1}{C^2}}{2ln(C)}}$ sounds reasonable since it decreases when the number of classes ($C$) increases. We refer to the hard thresholding variant utilizing this implicit threshold as **EO-H**.
>
> Here we conduct a simple comparison on the first domain (Ar) of the OfficeHome dataset and show the results across different category shifts below (**EO** denotes the objective in Eq. (3)).
>
> | Metrics (%) | ACC (C) | AUC (C) | ACC (P) | AUC (P) | ACC (O) | AUC (O) | ACC (OP) | AUC (OP) | ACC (Avg.) | AUC (Avg.) |
> |---------|---------|---------|---------|---------|---------|---------|---------|---------|---------|---------|
> | CLIP    | 70.1 | 74.9 | 70.1 | 74.9 | 70.2 | 74.9 | 70.1 | 74.9 | 70.1 | 74.9 |
> | EO-ID   | 69.9 | 75.7 | 69.8 | 75.6 | 70.4 | 76.3 | 70.4 | 76.3 | 70.1 | 76.0 |
> | EO-H    | 68.8 | 74.7 | 68.8 | 74.4 | 69.6 | 76.2 | 70.1 | 76.0 | 69.3 | 75.3 |
> | EO      | 68.9 | 74.9 | 68.6 | 74.7 | 69.6 | 75.8 | 69.5 | 75.7 | 69.1 | 75.3 |
> | UEO     | 72.3 | 75.1 | 72.1 | 75.7 | 73.2 | 74.0 | 72.3 | 74.9 | 72.5 | 74.9 |
>
> [(C): closed-set, (P): partial-set, (O): open-set, (OP): open-partial]
>
> As can be seen from the table above, UEO consistently outperforms all three thresholding methods in terms of ACC while being inferior to them in terms of AUC under open-set and open-partial shifts. Notably, **the thresholding technique proves harmful for closed-set and partial-set shifts** where no OOD samples exist in the unlabeled data. In summary, UEO demonstrates superior suitability compared to these thresholding variants.

---

> > ### Comment · Reviewer_eDVD · 2023-11-22
> > **Thanks for adding the comparison with different thresholding strategies**
> >
> > I would thank the authors for adding the comparison with other thresholding strategies and also for correcting my on the derivation of the implicit threshold when $w$ follows a uniform distribution.
> >
> > Regarding the above table provided by the authors, it is interesting to find that hard-thresholding strategies like EO-H and EO-ID perform on par with Eq. (3). Does this imply that Eq. (3) is less effective in adaptively reweighting ID and OOD samples? Furthermore, as Eq. (4) is a modified version of Eq. (3) with catering for the case that no OOD samples are presented in the minibatch, is it possible to achieve the same goal without starting from Eq. (3)? That is, what I concern is that the derivation from Eq. (3) to Eq. (4) is less convincing given the above results in the table.

---

> > > ### Author Response · Authors · 2023-11-23
> > > **End of discussion approaching**
> > >
> > > Dear Reviewer,
> > >
> > > Since the discussion deadline is approaching in less than 12 hours, we kindly request your feedback on whether the follow-up response adequately addresses your concerns. If you have any more questions, we would be happy to provide further clarification.
> > >
> > > Your timely response is greatly appreciated.
> > >
> > > Thanks.

---

> ### Author Response · Authors · 2023-11-16
>
> > **[Q5]**. The proposed method appears to be a general OOD method that can be also applied to traditional classification networks. I wonder why this method is applied to only CLIP method instead of extending it to other pre-trained backbones.
>
> **[A5]**. Indeed, our objective in Eq. (4) is adaptable to traditional classification networks as well. The reason why we focus on CLIP lies in its flexibility, as a pre-trained model can be easily acquired based on class names without the need for labeled data, rendering it a versatile framework.
>
> Regarding the traditional classification networks, we employ the ResNet-50 backbone with the [backbone - bottleneck layer - classifier layer] architecture to train the source model. During unsupervised fine-tuning in the target domain, we only update the parameters of the bottleneck layer (parameter-efficient), the results on two different tasks are shown below. On both tasks, UEO significantly outperforms other methods in terms of both ACC and AUC metrics.
>
> | OF (D->A) | ACC (C) | AUC (C) | ACC (P) | AUC (P) | ACC (O) | AUC (O) | ACC (OP) | AUC (OP) | ACC (Avg.) | AUC (Avg.) |
> |---------|---------|---------|---------|---------|---------|---------|---------|---------|---------|---------|
> | source only   | 65.2 | 71.8 | 65.2 | 71.8 | 65.2 | 71.8 | 65.2 | 71.8 | 65.2 | 71.8 |
> | UPL        | 69.7 | 73.8 | 67.7 | 72.2 | 69.9 | 74.0 | 68.1 | 70.0 | 68.8 | 72.5 |
> | POUF       | 69.9 | 75.7 | 68.9 | 73.7 | 70.1 | 74.2 | 68.5 | 71.7 | 69.4 | 73.8 |
> | DANCE      | 65.0 | 72.5 | 66.9 | 73.5 | 65.8 | 73.2 | 64.8 | 71.8 | 65.6 | 72.7 |
> | EntMin     | 64.3 | 72.0 | 64.9 | 72.8 | 63.2 | 69.8 | 63.9 | 70.5 | 64.1 | 71.3 |
> | InfoMax    | 71.8 | 77.1 | 70.1 | 74.5 | 72.2 | 74.2 | 69.4 | 70.8 | 70.9 | 74.2 |
> | UEO        | 71.6 | 75.5 | 71.1 | 74.9 | 71.6 | 75.0 | 70.2 | 74.1 | 71.1 | 74.9 |
>
> | OH (Pr->Ar) | ACC (C) | AUC (C) | ACC (P) | AUC (P) | ACC (O) | AUC (O) | ACC (OP) | AUC (OP) | ACC (Avg.) | AUC (Avg.) |
> |---------|---------|---------|---------|---------|---------|---------|---------|---------|---------|---------|
> | source only   | 54.6 | 64.5 | 54.5 | 64.5 | 54.6 | 64.5 | 54.5 | 64.5 | 54.6 | 64.5 |
> | UPL        | 57.0 | 65.3 | 55.6 | 65.8 | 56.2 | 65.7 | 55.4 | 64.9 | 56.1 | 65.4 |
> | POUF       | 57.1 | 64.6 | 56.1 | 64.6 | 57.2 | 63.6 | 56.4 | 63.4 | 56.7 | 64.0 |
> | DANCE      | 53.9 | 65.2 | 54.2 | 65.0 | 53.2 | 62.7 | 54.0 | 64.7 | 53.8 | 64.4 |
> | EntMin     | 52.5 | 64.3 | 53.2 | 64.4 | 51.5 | 62.6 | 52.1 | 62.4 | 52.3 | 63.4 |
> | InfoMax    | 57.9 | 65.4 | 56.9 | 65.5 | 57.2 | 63.8 | 57.1 | 63.6 | 57.3 | 64.5 |
> | UEO        | 58.3 | 66.3 | 58.2 | 66.2 | 57.9 | 64.9 | 58.2 | 64.7 | 58.1 | 65.5 |

---

> ### Author Response · Authors · 2023-11-21
> **Invitation to further discussion**
>
> Dear reviewer,
>
> We genuinely appreciate the time and effort you've invested in reviewing our paper. We have carefully provided relevant responses and results to your concerns. We are eager to further discuss with you and gain your insights **before the end of the Author/Reviewer phase**. Please let us know if any aspect of our work remains unclear or if you have additional feedback.
>
> Thank you.

---

> ### Author Response · Authors · 2023-11-22
> **Further clarification on the derivation from Eq. (3) to Eq. (4)**
>
> We express our sincere gratitude to the reviewer for providing us with the opportunity to address the remaining concerns during the author/reviewer phase.
>
> To validate the effectiveness of both the weighting strategy in Eq. (3) and the relaxation (derivation) strategy in Eq. (4), we extend our comparisons to three additional domains within the OfficeHome dataset. We present the averaged results across various category shifts **[(C): closed-set, (P): partial-set, (O): open-set, (OP): open-partial]** and domains **[OH (Ar)/ OH (Cl)/ OH (Pr)/ OH (Re)]** for different variants in the table below. Note that, UEO-H/ID replaces $\mathbb{E}(H(p(x)))$ in EO-H/ID with $H(\mathbb{E}(p(x)))$.
>
> | OH (Avg.)|  ACC (Avg.) | AUC (Avg.) |
> |---------|---------|---------|
> | CLIP   | 73.2 | 75.0 |
> | EO-ID  | 72.9 | 75.7 |
> | EO-H   | 72.3 | 75.5 |
> | EO     | 72.4 | 75.4 |
> | UEO-ID | 74.5 | 75.7 |
> | UEO-H  | 74.8 | 75.9 |
> | UEO    | 76.6 | 75.6 |
>
> We can derive three main conclusions:
> - Although the weighting strategy does not enable EO to surpass EO-H in Eq.(3), comparing UEO with UEO-H demonstrates the effectiveness of the weighting strategy in the final objective.
> - The relaxation (derivation) strategy significantly improves all three variants (EO-ID, EO-H, and EO), with a notable impact on the ACC metric.
> - UEO achieves the best performance while considering both ACC and AUC metrics.
>
> &nbsp;
>
> We attach the detailed results under each category shift and each domain as follows.
>
>  [(C): closed-set, (P): partial-set, (O): open-set, (OP): open-partial]
>
> &nbsp;
>
> | OH (Ar)| ACC (C) | AUC (C) | ACC (P) | AUC (P) | ACC (O) | AUC (O) | ACC (OP) | AUC (OP) | ACC (Avg.) | AUC (Avg.) |
> |---------|---------|---------|---------|---------|---------|---------|---------|---------|---------|---------|
> | CLIP   | 70.1 | 74.9 | 70.1 | 74.9 | 70.2 | 74.9 | 70.1 | 74.9 | 70.1 | 74.9 |
> | EO-ID  | 69.9 | 75.7 | 69.8 | 75.6 | 70.4 | 76.3 | 70.4 | 76.3 | 70.1 | 76.0 |
> | EO-H   | 68.8 | 74.7 | 68.8 | 74.4 | 69.6 | 76.2 | 70.1 | 76.0 | 69.3 | 75.3 |
> | EO     | 68.9 | 74.9 | 68.6 | 74.7 | 69.6 | 75.8 | 69.5 | 75.7 | 69.1 | 75.3 |
> | UEO-ID | 70.7 | 75.7 | 70.1 | 75.6 | 71.3 | 75.7 | 71.6 | 75.4 | 70.9 | 75.6 |
> | UEO-H  | 71.2 | 75.4 | 70.9 | 75.2 | 72.0 | 75.6 | 71.8 | 75.3 | 71.5 | 75.4 |
> | UEO    | 72.3 | 75.1 | 72.1 | 75.7 | 73.2 | 74.0 | 72.3 | 74.9 | 72.5 | 74.9 |
>
> &nbsp;
>
> | OH (Cl)| ACC (C) | AUC (C) | ACC (P) | AUC (P) | ACC (O) | AUC (O) | ACC (OP) | AUC (OP) | ACC (Avg.) | AUC (Avg.) |
> |---------|---------|---------|---------|---------|---------|---------|---------|---------|---------|---------|
> | CLIP   | 55.3 | 64.7 | 55.2 | 64.7 | 55.2 | 64.7 | 55.3 | 64.7 | 55.3 | 64.7 |
> | EO-ID  | 56.9 | 65.9 | 55.8 | 65.5 | 57.1 | 63.5 | 56.3 | 64.8 | 56.5 | 64.9 |
> | EO-H   | 56.0 | 65.4 | 54.8 | 65.7 | 55.9 | 65.5 | 55.7 | 65.9 | 55.6 | 65.6 |
> | EO     | 56.4 | 65.5 | 55.7 | 65.7 | 56.5 | 65.5 | 55.7 | 66.0 | 56.1 | 65.7 |
> | UEO-ID | 59.9 | 66.3 | 58.2 | 66.4 | 60.0 | 65.0 | 58.8 | 64.1 | 59.2 | 65.5 |
> | UEO-H  | 59.8 | 66.7 | 59.6 | 66.4 | 60.4 | 66.3 | 59.7 | 66.4 | 59.9 | 66.4 |
> | UEO    | 61.2 | 64.8 | 61.4 | 65.2 | 62.0 | 64.7 | 61.0 | 64.9 | 61.4 | 64.9 |
>
> &nbsp;
>
> | OH (Pr)| ACC (C) | AUC (C) | ACC (P) | AUC (P) | ACC (O) | AUC (O) | ACC (OP) | AUC (OP) | ACC (Avg.) | AUC (Avg.) |
> |---------|---------|---------|---------|---------|---------|---------|---------|---------|---------|---------|
> | CLIP   | 84.2 | 78.0 | 84.2 | 78.0 | 84.2 | 78.0 | 84.2 | 78.0 | 84.2 | 78.0 |
> | EO-ID  | 81.6 | 77.6 | 81.0 | 76.0 | 83.0 | 79.7 | 83.1 | 79.0 | 82.2 | 78.1 |
> | EO-H   | 82.5 | 77.2 | 79.1 | 74.4 | 82.6 | 79.4 | 83.1 | 78.8 | 81.8 | 77.4 |
> | EO     | 82.3 | 77.1 | 81.2 | 75.9 | 82.7 | 79.3 | 83.0 | 78.8 | 82.3 | 77.8 |
> | UEO-ID | 83.8 | 78.0 | 82.7 | 77.7 | 84.6 | 78.8 | 84.8 | 78.5 | 84.0 | 78.2 |
> | UEO-H  | 83.5 | 78.0 | 83.1 | 77.3 | 84.8 | 78.8 | 85.0 | 78.7 | 84.1 | 78.2 |
> | UEO    | 87.7 | 79.6 | 87.0 | 80.3 | 87.0 | 78.5 | 86.5 | 78.9 | 87.1 | 79.3 |
>
> &nbsp;
>
> | OH (Re)| ACC (C) | AUC (C) | ACC (P) | AUC (P) | ACC (O) | AUC (O) | ACC (OP) | AUC (OP) | ACC (Avg.) | AUC (Avg.) |
> |---------|---------|---------|---------|---------|---------|---------|---------|---------|---------|---------|
> | CLIP   | 83.0 | 82.5 | 83.1 | 82.5 | 83.1 | 82.5 | 83.0 | 82.5 | 83.0 | 82.5 |
> | EO-ID  | 82.2 | 83.2 | 82.7 | 82.8 | 82.9 | 84.5 | 83.5 | 84.5 | 82.8 | 83.8 |
> | EO-H   | 81.6 | 82.1 | 82.1 | 82.5 | 82.9 | 84.8 | 83.4 | 84.8 | 82.5 | 83.5 |
> | EO     | 81.0 | 82.1 | 82.0 | 81.7 | 82.7 | 83.9 | 82.8 | 84.0 | 82.1 | 83.0 |
> | UEO-ID | 83.1 | 83.1 | 83.5 | 83.0 | 84.2 | 84.2 | 84.6 | 84.1 | 83.8 | 83.6 |
> | UEO-H  | 83.0 | 82.8 | 83.2 | 83.3 | 84.1 | 84.3 | 84.5 | 84.1 | 83.7 | 83.6 |
> | UEO    | 85.6 | 82.9 | 85.4 | 83.9 | 85.3 | 82.6 | 85.5 | 83.5 | 85.4 | 83.2 |

---

> ### Author Response · Authors · 2023-11-23
> **Your Feedback is Appreciated in the Last Minute.**
>
> Dear reviewer,
>
> Concerning the only issue raised about **the convincing derivation from Eq. (3) to Eq. (4)**, we have presented comprehensive comparisons between UEO and various variants in the rebuttal phase. We are eager to understand if these results have successfully addressed your reservations regarding **the effectiveness of both the weighting strategy and the relaxation strategy in Eq. (4)**.
>
> As for the other concerns that were not reiterated in your recent response, we would appreciate confirmation on whether they have been satisfactorily addressed in our previous response. **We would be very grateful if you could provide your feedback before the end of the rebuttal phase.**
>
> The authors

---

### Meta-Review · Area_Chair_VoXG · 2023-12-19

**Metareview:**

This paper studies a new setting of fine-tuning of vision-language models (focusing on CLIP) with samples containing unknown classes which have both overlapping and non-overlapping classes. The paper proposes an entropy-based objective, with entropy minimization on ID samples and maximization on OOD samples. Results are shown across a range of datasets such as Office, OfficeHome, DomainNet, etc.

  While the reviewers appreciated the interestingness of the setting, they raised a number of significant concerns, several of which were shared across multiple reviewers. This includes: 1) Novelty of the method given that entropy minimization/maximization is a common technique across a range of ML sub-fields (eDVD, pTa8), 2) Lack of clarity in the theoretical motivation (eDVD), 3) Missing ablations or need for additional datasets (pTa8), and 4) Lack of significance of the results compared to baselines in several cases (spW5, tymK). While the authors provided a rebuttal addressing some of these concerns, especially additional datasets, the reviewers expressed lingering concerns including about the similarity of the method to prior works and lack of better results in several cases. Indeed, the changed scores are still below the acceptance recommendation, and the strongest reviewer also had the least confidence and least specific questions.

  Having considered the paper, the reviews, the rebuttal, and discussion, I agree that this paper does not pass the acceptance threshold. While the new setting may be interesting and I encourage the authors to continue to pursue it, the method proposed both is not scientifically significant since it does not propose a significantly new method and does not convincingly perform better than the baselines in cases where it should. I encourage the authors to significantly bolster these contributions in future submissions.

**Justification For Why Not Higher Score:**

Overall, the mean score is biased by the positive reviewer that was least confident and only had generic questions, and the other two reviewers still expressed concerns. After looking over the paper, I share the concerns about the strength of both the algorithmic and empirical contributions.

**Justification For Why Not Lower Score:**

N/A

---

### Decision · Program_Chairs · 2024-01-16

Reject